# The GTPase IFT27 is involved in both anterograde and retrograde intraflagellar transport

**Diego Huet[1,2], Thierry Blisnick[1], Sylvie Perrot[1], Philippe Bastin[1]\***

[1]Trypanosome Cell Biology Unit, Institut Pasteur & CNRS, Paris, France; [2]Université Pierre et Marie Curie, Cellule Pasteur-UPMC, Paris, France

**Abstract** The construction of cilia and flagella depends on intraflagellar transport (IFT), the bidirectional movement of two protein complexes (IFT-A and IFT-B) driven by specific kinesin and dynein motors. IFT-B and kinesin are associated to anterograde transport whereas IFT-A and dynein participate to retrograde transport. Surprisingly, the small GTPase IFT27, a member of the IFT-B complex, turns out to be essential for retrograde cargo transport in *Trypanosoma brucei*. We reveal that this is due to failure to import both the IFT-A complex and the IFT dynein into the flagellar compartment. To get further molecular insight about the role of IFT27, GDP- or GTP-locked versions were expressed in presence or absence of endogenous IFT27. The GDP-locked version is unable to enter the flagellum and to interact with other IFT-B proteins and its sole expression prevents flagellum formation. These findings demonstrate that a GTPase-competent IFT27 is required for association to the IFT complex and that IFT27 plays a role in the cargo loading of the retrograde transport machinery.

## Introduction

Cilia and flagella (interchangeable terms) are microtubule-based organelles surrounded by a specialized plasma membrane. These organelles are found protruding at the surface of a wide range of eukaryotic cells where they exert several roles including cellular motility, sensory reception and developmental signaling (*Fliegauf et al., 2007*; *Drummond, 2012*). Although cilia and flagella can differ by their functions, they share the same basic structure and are mainly assembled by an evolutionary conserved process called intraflagellar transport (IFT) (*Kozminski et al., 1993*; *Rosenbaum and Witman, 2002*). During IFT, large protein complexes called IFT trains are moved from the base to the tip of the flagellum by kinesin-2 (anterograde IFT) (*Kozminski et al., 1995*) and back towards the base by cytoplasmic dynein-1b/2 (retrograde IFT) (*Pazour et al., 1998*). The IFT particles are composed of at least 20 different proteins (*Cole et al., 1998*; *Follit et al., 2009*; *Taschner et al., 2011*) and can be divided in two different complexes known as complex A (made of at least 6 proteins) and B (made of at least 14 proteins) that are needed for retrograde and anterograde IFT respectively. Inactivation of IFT-B proteins or of the kinesin motor results in the failure to construct the flagellum whereas IFT-A proteins and the IFT dynein contribute to retrograde transport. Nevertheless, little is known about how the two processes are interconnected, as the interaction between the IFT complexes and their molecular motors is still poorly understood.

Structurally, IFT proteins contain several protein–protein interaction domains such as WD40, tetratricopeptide repeats, β-propeller or coiled-coil domains (*Cole, 2003*). However, two IFT proteins belonging to the IFT complex B do not share those features. Instead, they exhibit sequence homologies with the family of Rab-like small G proteins, which can function to regulate vesicle movement along actin and tubulin networks (*Stenmark, 2009*). This suggests that they could be involved in IFT regulation. The first one, IFT22 (or Rab-like 5) traffics inside the sensory neurons of *Caenorhabditis*

\*For correspondence: pbastin@pasteur.fr

**Competing interests:** The authors declare that no competing interests exist.

**eLife digest** Long, thin structures called cilia and flagella are found on the surface of many cells, and perform a range of roles, including propelling the cells around or sensing changes in the surrounding environment. A process called intraflagellar transport (IFT for short) is responsible for flagellum construction in eukaryotic cells. Protein complexes called IFT trains carry the building blocks that make up flagella along microtubule 'tracks' between the base and the tip of a flagellum.

IFT trains are made from two different protein complexes called IFT-A and IFT-B, which are dragged by various molecular motors. The IFT-B complex is necessary for the train to move towards the tip of the flagellum, and so enables the flagellum to grow. The IFT-A protein complex is required to recycle the train back towards the base of the flagellum.

Huet et al. examined the role that a protein called IFT27 plays in intraflagellar transport. IFT27 is part of the IFT-B complex, and so it was thought to only affect how flagella grow. However, short flagella still grow when IFT27 is absent, but they are filled with IFT trains that are not able to reverse back from the tip. Huet et al. reveal that the IFT-A complex and the molecular motor that is essential for reversing the train are not transported into the flagellum if IFT27 is not present. This is therefore an unusual case of an IFT-B protein affecting the IFT-A complex and the transport back to the base.

IFT27 also affects how the IFT-B complex forms. ITF27 can bind to some small molecules, which can switch the protein 'on' or 'off'. Huet et al. found that when IFT27 is switched off it is not transported into flagella, and also cannot bind to some of the other proteins in the IFT-B complex. This means that if IFT27 is locked in an inactive state, the IFT-B complex does not form, and a flagellum cannot grow. Therefore, activated IFT27 is needed for putting together the IFT train and to ensure its movement in either direction along the microtubule tracks.

*elegans*, where its disruption does not affect cilia assembly but produces an intriguing extended lifespan phenotype (*Schafer et al., 2006*). IFT22 also undergoes IFT in the flagellum of the protist *Trypanosoma brucei* but surprisingly knocking down the protein resulted in severe flagellar assembly defects, typical of inhibition of retrograde transport (*Adhiambo et al., 2009*). In *Chlamydomonas*, IFT22 regulates the availability of IFT proteins inside the cell (*Silva et al., 2012*). Despite being homologous to the Rab-like proteins, IFT22 lacks the G4 domain needed for recognition of the guanine base (*Paduch et al., 2001*) and GTP binding and hydrolysis has not been demonstrated to date. Therefore, IFT22 might have diverged from its ancestral GTPase function to complete specific roles in *C. elegans*, *T. brucei*, and *Chlamydomonas*.

The second protein is termed IFT27 (or Rab-like 4) and contrary to IFT22, it possesses all five RAS-GTPase consensus sequences needed for GTPase activity and GDP binding. Structurally, the protein is similar to Rab8 and Rab11 and is capable of GTP hydrolysis (*Bhogaraju et al., 2011*). IFT27 is part of the IFT complex B core, where the protein interacts directly with IFT25 (*Lucker et al., 2005*). This IFT25/27 sub-complex is associated with the rest of the IFT complex B prior to flagellum entry, suggesting a possible regulatory role during the entry of the IFT machinery (*Wang et al., 2009*). In *Chlamydomonas*, RNAi-mediated knockdown of IFT27 causes flagellar defects (*Qin et al., 2007*), supporting its role during flagellar assembly, although its exact contribution was not investigated. In this study, we analyzed the function of IFT27 in *T. brucei*, the parasite that causes sleeping sickness and that is also an amenable model to study flagellar assembly (*Kohl and Bastin, 2005*; *Ralston and Hill, 2008*) as well as the IFT components (*Absalon et al., 2008b*; *Adhiambo et al., 2009*; *Franklin and Ullu, 2010*; *Bhogaraju et al., 2013*; *Buisson et al., 2013*). We found that IFT27 travels by IFT and associates with other IFT-B proteins. RNAi knockdown surprisingly produced short flagella filled with IFT-like material and axoneme assembly defects. This phenotype is explained by the absence of both the IFT140 protein (a member of the IFT-A complex) and the IFT dynein motor from the flagellar compartment. Generation of constitutively active and inactive forms of IFT27 produced further insights: while expression of the active version of the protein complements the RNAi phenotype, this was not the case of the inactive version that was unable to penetrate the flagellar compartment. Surprisingly, its expression in the absence of endogenous protein led to the formation of short flagella that do not accumulate IFT-like material. This inactive version is unable to interact with two other IFT-B proteins, suggesting that IFT27 must be in a GTP-bound state in order to interact with the B-complex and enter

the flagellum. These results show that IFT27, an IFT-B protein, performs two separate functions: one in the import of both the IFT-A complex and IFT dynein motors and one in the assembly of the IFT-B complex.

## Results

### IFT27 encodes a putative Rab-like protein

The *T. brucei IFT27* gene (TritrypDB Accession number Tb927.3.5550) has a 552 nucleotide-long sequence that encodes a predicted protein of 183 amino acids (predicted molecular weight of 20.64 kDa). BLAST analyses show that IFT27 shares significant homology (E value = 2e$^{-27}$) with the Rab-like 4 (RABL4) GTPase found in *Homo sapiens* and vertebrates. *IFT27* homologues are present in the genomes of all ciliated organisms except in *Drosophila, C. elegans, Giardia,* and some ferns and mosses (*van Dam et al., 2013*). The predicted trypanosome protein contains all five consensus domains needed for GTP/GDP binding and GTPase activity found in most Rab proteins (*Figure 1*), indicating that IFT27 could be a functional small G protein. Additionally, all IFT27 sequences lack the prenylation motif (two cysteins at the C terminal end) found in Rab proteins suggesting that the protein is not lipid modified and thus unlikely to interact with the cellular membrane.

### IFT27 traffics in the trypanosome flagellum

Two approaches were used to determine the location of IFT27 in *T. brucei*. First, the full-length protein was expressed and used to produce antisera in mice. Second, a GFP::IFT27 fusion protein was expressed in procyclic trypanosomes. Western blot analyses using the anti-IFT27 antibody showed a single band migrating at a position close to the predicted size of 20 kDa in wild-type cells (*Figure 2A*). In trypanosomes expressing the GFP::IFT27 fusion, the same antibody detected an additional band

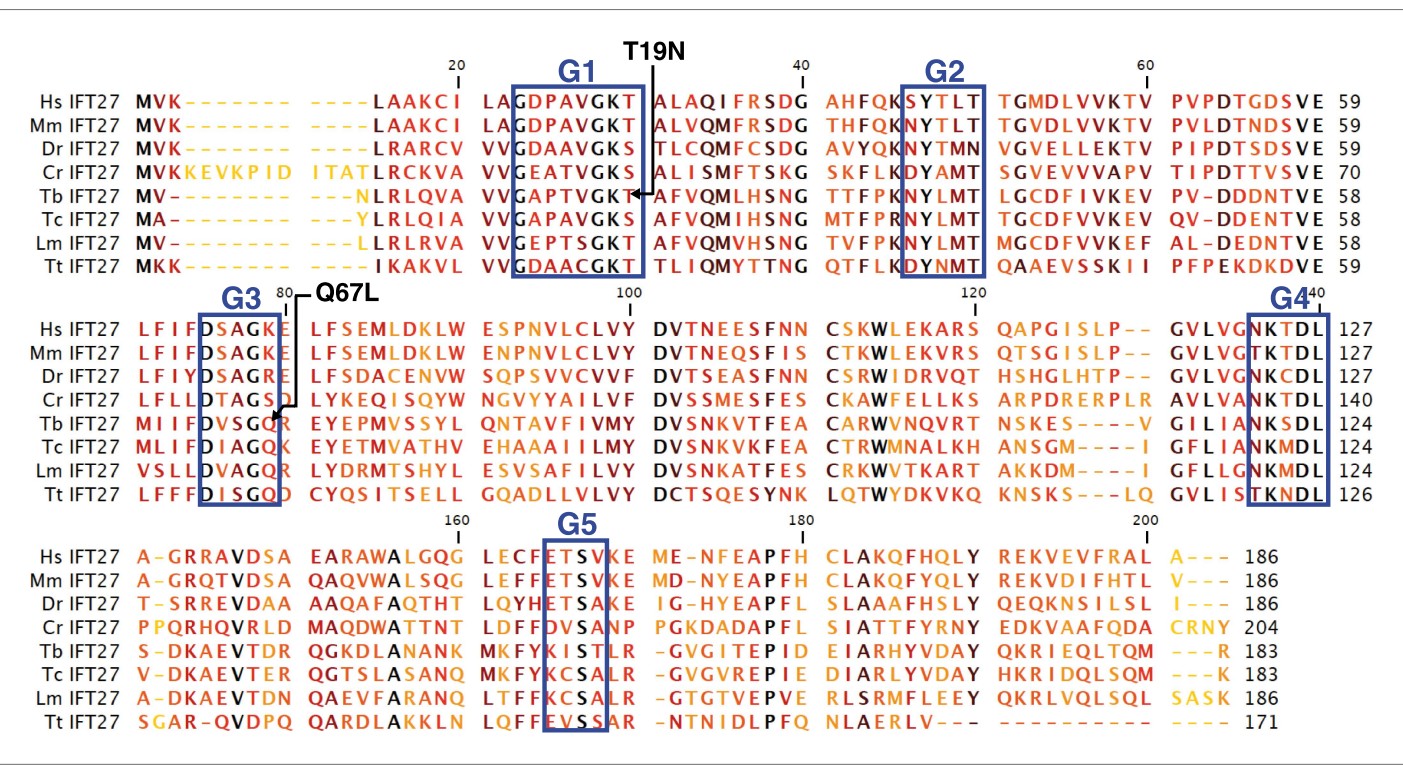

**Figure 1**. Sequence alignment of deduced amino acid sequences of IFT27 homologues and modified *IFT27* sequences. Alignment was generated using CLC main workbench; the most conserved residues are shown in black, the less conserved in yellow. G1–G5 indicates conserved motifs implicated in nucleotide binding domains and GTPase activity. Dashes indicate gaps introduced to optimize the alignment. Arrowheads indicate missense mutations created in *IFT27*$^{RNAiRES}$. Abbreviations and accession numbers are as follows: Hs: *Homo sapiens*, NP_006851.1, Mm: *Mus musculus*, NP_080207.1, Dr: *Danio rerio*, NP_001008588.1, Cr: *Chlamydomonas reinhardtii*, XP_001689745.1, Tb: *Trymanosoma brucei*, Tb927.3.5550, Tc: *Trypanosoma cruzi*, AY371275, Lm: *Leishmania major*, LmjF.29.0090, Tt: *Tetrahymena thermophila*, TTHERM_00298510.

migrating close to the 50 kDa marker. This molecular weight is compatible with the expected mass of the fusion protein (*Figure 2A*). The anti-IFT27 antibody was then used in immunofluorescence assays in combination with DAPI to stain both nuclear and mitochondrial DNA, the latter being an easy marker of basal body positioning in trypanosomes (*Robinson and Gull, 1991*). In wild-type cells, the anti-IFT27 antibody produced a signal all along the flagellum, starting at the base of the organelle and reaching its distal tip where it was sometimes brighter (*Figure 2B*). The GFP-fusion protein showed a similar localization, with the presence of the protein at the flagellum base and inside the flagellum. Co-staining of the GFP-tagged protein and IFT27 showed a clear colocalization inside the flagellum (*Figure 3A*) and the use of live microscopy demonstrated that the fusion protein clearly traffics along the flagellum (*Figure 2C,D*; *Video 1*) where typical bidirectional IFT was visualized. Similar IFT trafficking was observed for other IFT-related proteins in trypanosomes including GFP::IFT52 (*Absalon et al., 2008b*), GFP::IFT22 (*Adhiambo et al., 2009*), and YFP::IFT81 (*Bhogaraju et al., 2013*). Kymographs were made to quantify IFT train speed (*Figure 2D*). The mean anterograde velocity was 2.5 ± 0.68 µm/s (n = 234) and mean retrograde velocity was measured at 3.8 ± 1.5 µm/s (n = 269). These speed values are similar to those previously reported for GFP::IFT52 (*Buisson et al., 2013*). Overall, these results show that IFT27 distribution and movement are similar to the other IFT-B proteins studied to date in *T. brucei*.

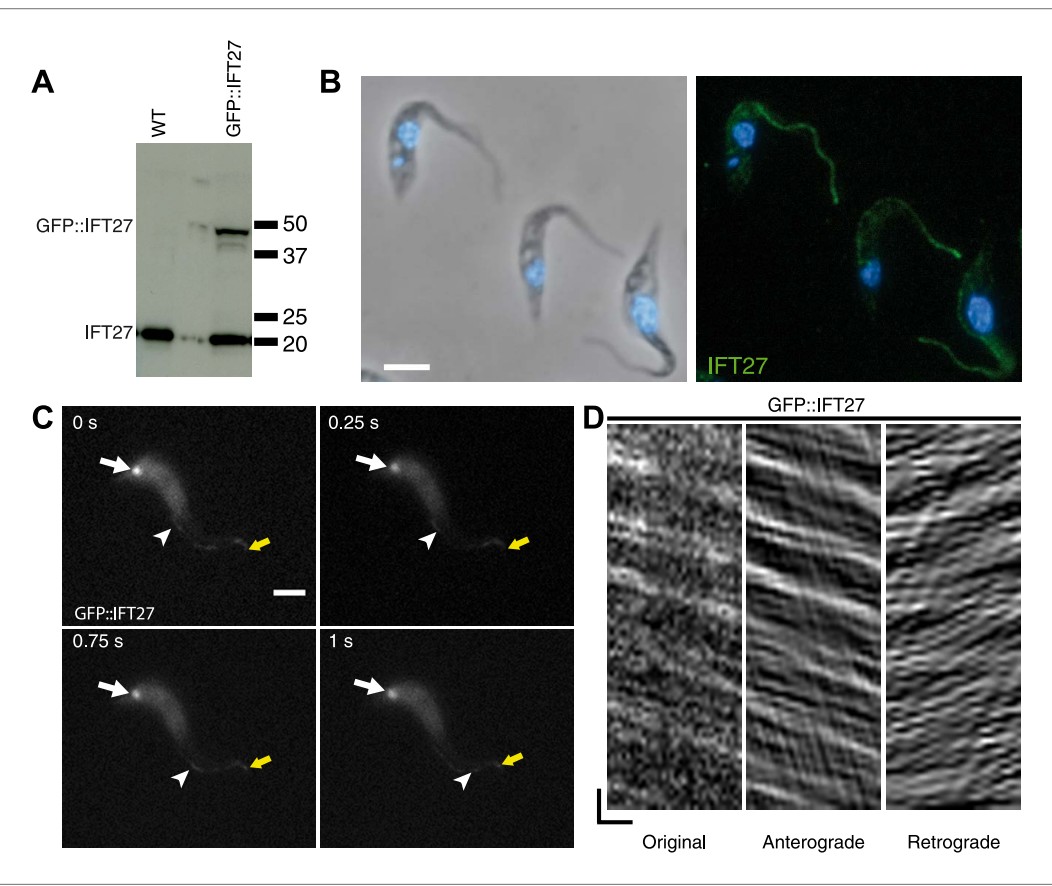

**Figure 2**. IFT27 is found inside the trypanosome flagellum where it travels by IFT. (**A**) Western blot of wild-type trypanosomes and cells expressing GFP::IFT27 probed with the anti-IFT27 polyclonal antibody. (**B**) Immunofluorescence of wild-type cells fixed in methanol using the anti-IFT27 polyclonal antibody. The first panel shows the phase-contrast image merged with DAPI staining and the second one shows the anti-IFT27 staining (green) merged with DAPI staining (blue). (**C**) Still images of a trypanosome expressing GFP::IFT27 (*Video 1*). The white arrow shows the fluorescent protein pool at the level of the flagellum base and the yellow arrow shows the distal tip of the flagellum. Arrowheads indicate the successive position of an anterograde IFT train. (**D**) Kymograph from *Video 1* shows clear IFT traces. Anterograde and retrograde events were separated as previously described (*Chenouard et al., 2010*, *Buisson et al., 2013*). Scale bars: 5 µm in **B** and **C**. In **D**, horizontal scale bar is 2 µm and vertical scale bar is 2 s.

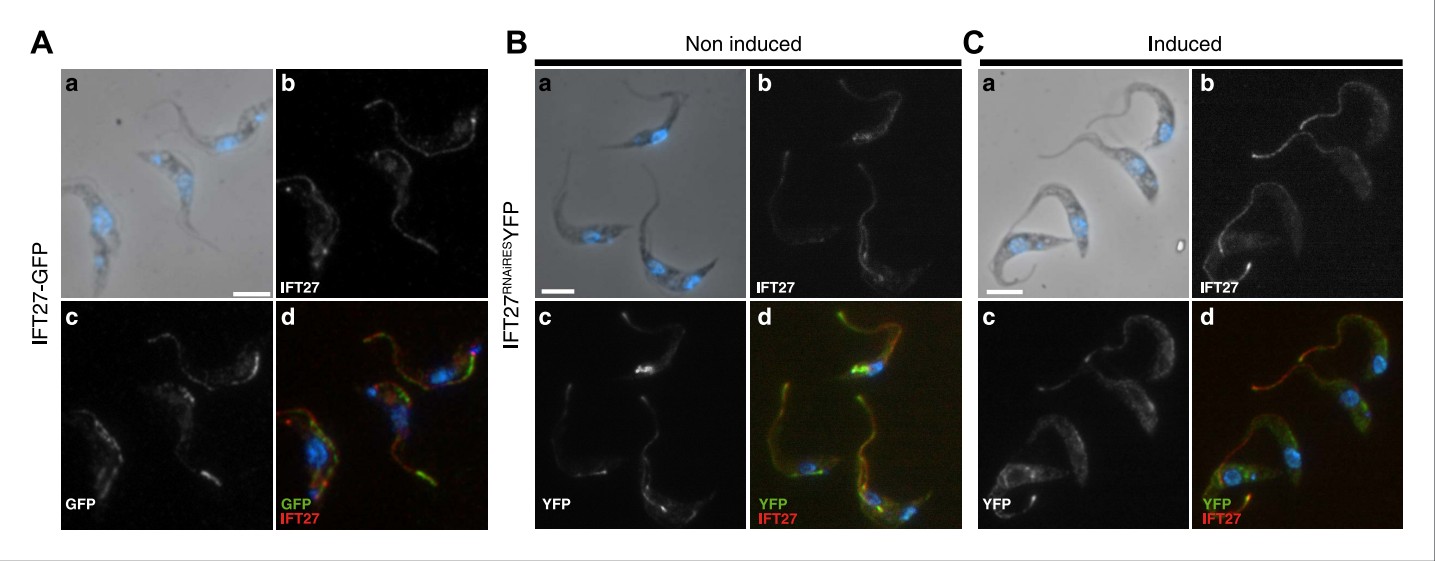

**Figure 3**. GFP::IFT27 and IFT27$^{RNAiRES}$::YFP colocalise with the endogenous IFT27. (**A**) Trypanosomes expressing GFP::IFT27 were fixed in methanol and stained simultaneously with the anti-IFT27 (red) antibody and the anti-GFP antibody (green). Top left and right panels show the phase-contrast image merged with DAPI staining and the anti-IFT27 staining respectively. Bottom left and right panels show the anti-GFP staining and the merged image. Non-induced (**B**) and 3-day induced (**C**) *IFT27$^{RNAiRES}$YFP* cells were fixed in methanol and immunofluorescence assays was performed using simultaneously the anti-IFT27 and the anti-GFP antibodies in order to locate the endogenous and the fluorescent version of IFT27. Immunofluorescence assays reveal that both proteins are found within the flagellum. Scale bars: 5 µm.

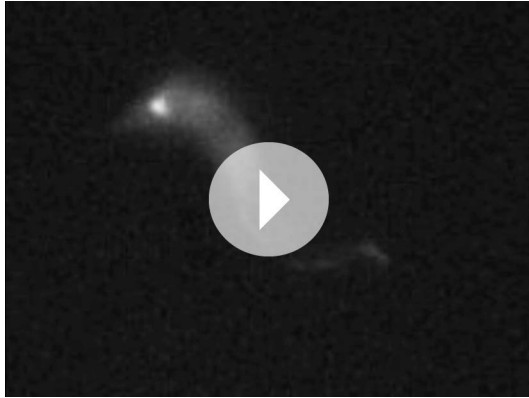

**Video 1**. IFT27 is found inside the trypanosome flagellum where it travels by IFT. Live procyclic, wild-type *T. brucei* cell transfected with GFP::IFT27 observed by time-lapse epifluorescence microscopy using a DMI4000 microscope at room temperature. Frames were taken every 250 ms for 30 s by an Evolve 512 EMCCD Camera. The resulting video was then exported to.AVI format with the ImageJ 1.47g13 software.

## IFT27 is required for cell growth and flagellum formation

To evaluate the role of IFT27, inducible RNA interference (RNAi) was used to deplete the expression of the protein in 29–13 procyclic parasites in culture. The 29-13 strain expresses the tetracycline repressor and the viral T7 RNA polymerase, which enables conditional expression of double stranded RNA leading to RNAi-mediated gene inhibition (*Wirtz et al., 1999*; *Wang et al., 2000*). Western blot analysis using the anti-IFT27 polyclonal antibody showed that the IFT27 expression was inhibited after 48 hr of induction (*Figure 4A*) and *IFT27$^{RNAi}$* cells showed a clear growth defect upon depletion of the protein (*Figure 4B*), in line with the essential role of the flagellum for the trypanosome cell cycle (*Kohl et al., 2003*). To analyze the impact of IFT27 depletion on flagellar formation, two markers were used: MAb25 (monoclonal, axoneme-specific) and L8C4 (monoclonal, paraflagellar rod-specific). Immunofluorescence assays results revealed that *IFT27$^{RNAi}$* cells induced for 72 hr assemble abnormally short flagella (*Figure 4C* bottom panel, arrowhead) with a mean axonemal length of 4.0 ± 6.8 µm (n = 100) instead of 20.5 ± 2.8 µm (n = 100) for non-induced controls (*Figure 4C*, top panel). Furthermore, L8C4 staining unpredictably displayed a very bright or very weak staining in the flagella of induced *IFT27$^{RNAi}$* cells, indicating that an excessive or a restrained amount of paraflagellar rod (PFR) material per unit length is found in the flagella of these cells (*Figure 4C*, bottom panel, arrow).

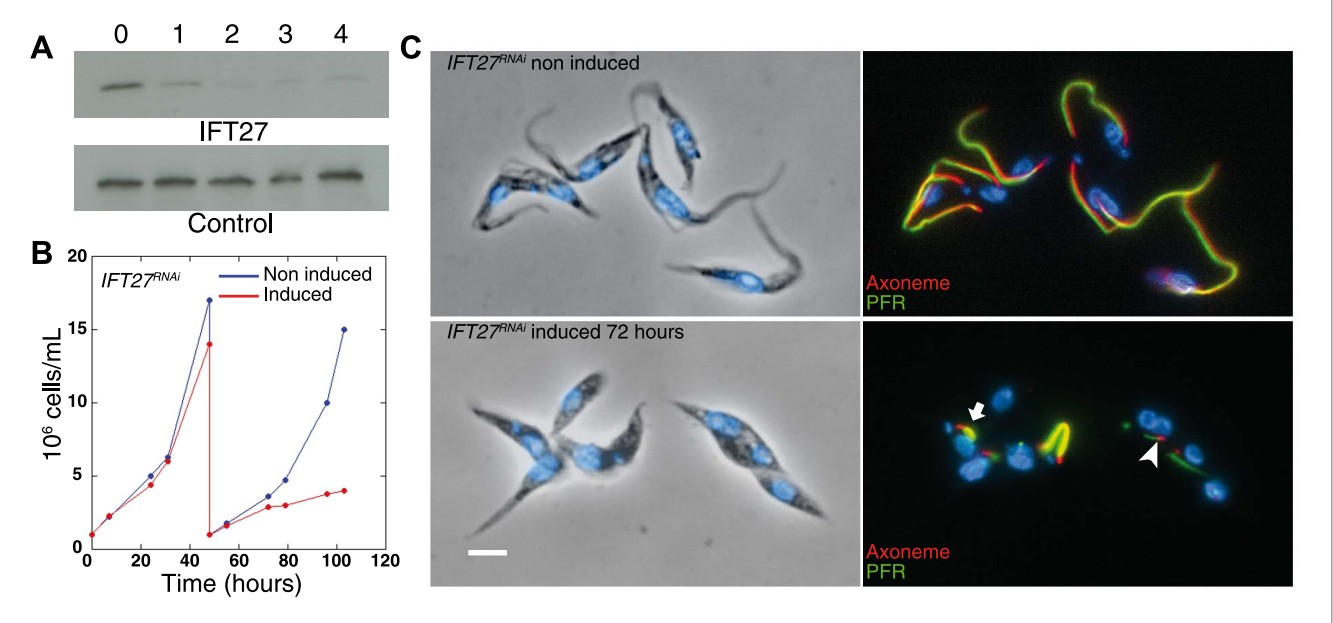

**Figure 4**. Silencing of IFT27 disrupts flagellum formation. (**A**) Western blot showing decrease of IFT27 protein upon RNAi silencing. Total protein samples of non-induced and induced cells were prepared after the indicated number of days. The membrane was incubated with the anti-IFT27 or the anti-aldolase as a control. (**B**) Growth curve of non-induced (blue) and induced (red) *IFT27^RNAi* cells. (**C**) Non-induced and 3-day induced *IFT27^RNAi* cells were fixed in methanol, stained with the Mab25 antibody to detect the axoneme (red) and the L8C4 antibody to detect the PFR (green) (right panels) then counterstained with DAPI (left panels). The arrowhead shows an abnormally short axoneme with an insufficient amount of PFR and the arrow indicates an excessive amount of PFR. Scale bar: 5 µm.

To evaluate whether and how IFT was affected, we first used immunofluorescence assays with three antibodies raised against different IFT-B proteins: IFT22 (*Adhiambo et al., 2009*), IFT172 (*Absalon et al., 2008b*), and PIFTC3 (*Franklin and Ullu, 2010*). IFT-B proteins in non-induced cells showed a uniform distribution along the flagellum (*Figure 5Aa–d*). However, unexpected accumulations of IFT172 and IFT22 were found in the short flagella of induced *IFT27^RNAi* cells (*Figure 5Ae–h*, arrows) and the same results were observed for PIFTC3, a candidate IFT-B protein (*Franklin and Ullu, 2010*) (unpublished data). In a second approach, the IFT-B core protein IFT52 was expressed as a GFP fusion protein (*Figure 5B*). Whereas in non-induced cells, GFP::IFT52 was localized in the cytoplasm but also concentrated at the base of the flagellum and inside the organelle (*Figure 5Ba–d*), 72-hr induced cells clearly showed an accumulation of the protein inside their short flagella (*Figure 5Be–h*). Together, these observations demonstrate that at least four members of the IFT-B complex accumulate inside the flagellum of the induced *IFT27^RNAi* cells.

The formation of short flagella filled with IFT proteins is typical of inhibition of retrograde IFT (*Pazour et al., 1999*; *Blacque et al., 2006*; *Absalon et al., 2008a*), suggesting that despite being a protein belonging to IFT-B complex, IFT27 could play a role in retrograde IFT. To further characterize this unexpected phenotype, transmission electron microscopy analyses were performed. Non-induced *IFT27^RNAi* cells looked like normal wild-type trypanosomes (*Figure 5Ca,b*), whereas induced cells exhibited major defects in the short flagellum. First, a significant accumulation of electron-dense material that likely corresponds to excessive IFT material was observed (*Figure 5Cc–d*). In addition, the axoneme was frequently disrupted and sometimes even split into two (*Figure 5C,D*) and an enlarged, often disorganized PFR structure was frequently observed. Over 61% of the flagellar sections (n = 77) displayed the combination of the three phenotypes. Other phenotypes included the presence of a membrane sleeve (18%) (*Davidge et al., 2006*; *Absalon et al., 2008b*), vesicles inside the lumen of the flagellar pocket (3%) and a loss of the alignment of the axonemal central pair relative to the PFR (14%) (*Branche et al., 2006*; *Gadelha et al., 2006*; *Ralston et al., 2006*; *Figure 6*).

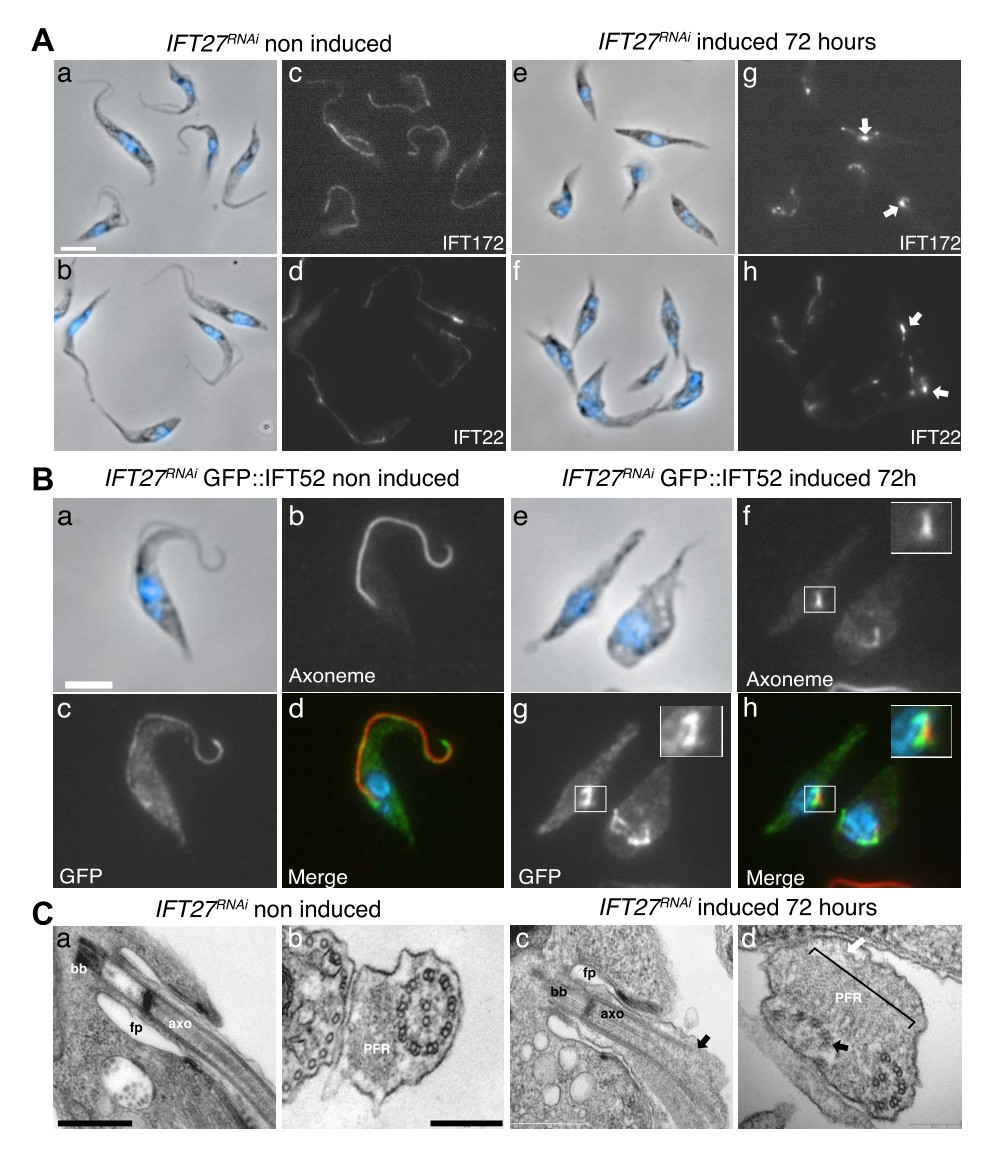

**Figure 5**. IFT complex B proteins accumulate in the short flagella of *IFT27^RNAi* cells. (**A**) *IFT27^RNAi* cells were non-induced (a to d) or induced for 3 days (e to h), labeled with the anti-IFT172 (a, c, e and g) or the anti-IFT22 antibody (b, d, f and h) and counterstained with DAPI (blue). Arrows show the accumulation of IFT proteins in the short flagella. (**B**) Non-induced (a to d) and 3-day induced (e to h) *IFT27^RNAi* *GFP-IFT52* cells were treated as above except that the anti-GFP was used. Insets in f and h show the axoneme and GFP::IFT52 localization respectively. Scale bar: 5 µm. (**C**) TEM of non-induced (a–b) and induced *IFT27^RNAi* cells (c–d). Black arrows show the axonemal defects, and the white arrow points to an excessive amount of IFT-like material. An enlarged PFR structure (bracket) is also visible. fp: flagellar pocket, axo: axoneme, bb: basal body. Scale bars: 500 nm in a and c, 200 nm in b and d.

## IFT27 contributes to the entry of the IFT dynein into the flagellum

To explain the surprising IFT retrograde phenotype, we examined the distribution of two elements involved in the retrograde transport machinery: the IFT dynein and IFT140, a member of the IFT-A complex. We generated an *IFT27^RNAi* cell line expressing the IFT dynein heavy chain (DHC1b) fused to GFP, and its localization was analyzed by immunofluorescence assays using the Mab25 and the anti-GFP antibodies (*Figure 7*). In non-induced *IFT27^RNAi* cells, the GFP::DHC1b signal was present around the base of the flagellum and displayed a punctuated staining inside the organelle and in the cyto-plasm (*Figure 7a–d*) confirming previous localization experiments performed with the anti-DHC1b

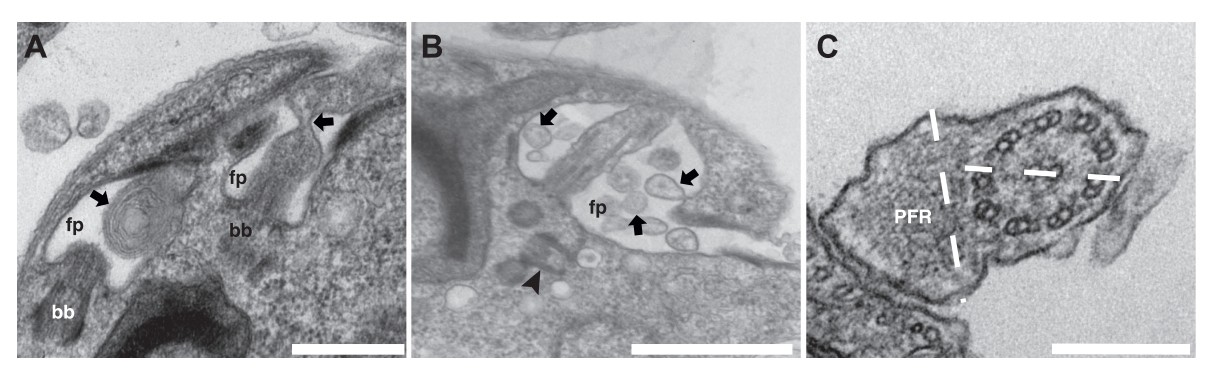

**Figure 6**. Additional phenotypes observed in the *IFT27^RNAi* cell line. Transmission electron micrographs of *IFT27^RNAi* cells induced for 3 days displaying different flagellar defects. (**A**) Membrane-like material in the flagellar pocket lumen and the presence of a flagellum 'sleeve' (arrows). (**B**) Vesicle-like material inside the flagellar pocket (arrows) and misorientation of the basal body (arrowhead). (**C**) Defect in central pair alignment relative to the PFR. Scale bars 500 nm in **A**, 1 μm in **B** and 200 nm in **C**.

antibody (our unpublished data). When *IFT27^RNAi* cells were induced for 3 days, DHC1b was not detected inside the short flagella but instead the protein appeared concentrated at the base of the flagellum (*Figure 7e–h*), suggesting defects in flagellum entry.

To determine the localization of the IFT-A complex, the core component IFT140 protein was fused to the Tandem Tomato (TdT) fluorescent protein upon endogenous tagging and expressed in *IFT27^RNAi* cells (*Figure 8*). In vivo observations confirmed that the fusion protein indeed traffics inside the flagellum (unpublished data) and its localization was then analyzed by immunofluorescence assays using an anti-DsRed antibody. Slides were stained simultaneously with the Mab25 and anti-IFT172 antibodies to localize the axoneme and the IFT-B complex respectively. In non-induced cells, the fusion protein is localized inside the flagellum compartment (*Figure 8b*) as for the anti-IFT172 (*Figure 8c*). In 3-day-induced *IFT27^RNAi* cells, TdT::IFT140 is only found concentrated at the base of the organelle and not inside the flagellar compartment (*Figure 8f,h*). In contrast, IFT172 is still aberrantly accumulated inside the organelle (*Figure 8g,h*) as observed previously.

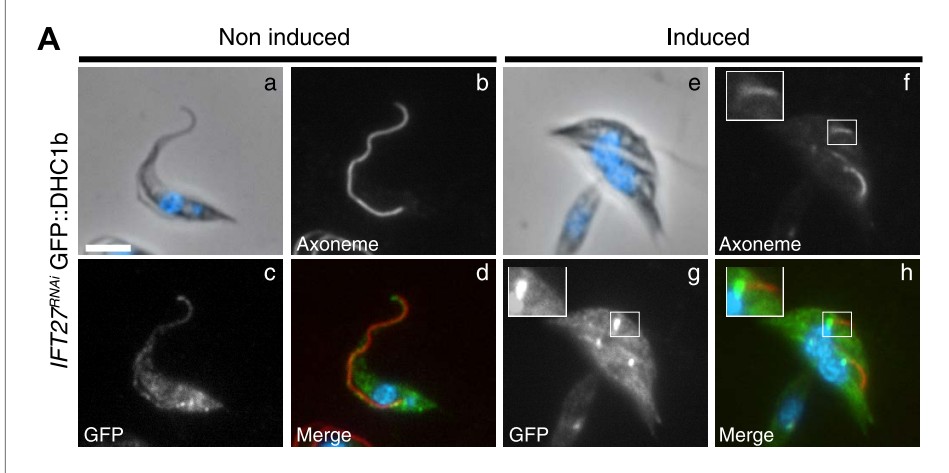

**Figure 7**. The IFT dynein is not able to access the flagellar compartment in the absence of IFT27. (**A**) Immunofluorescence of non-induced (a to d) and 3-day induced (e to h) *IFT27^RNAi GFP::DHC1b* cells fixed in methanol, counterstained with DAPI and stained with the Mab25 and anti-GFP antibodies. Insets in f and h show the localization of the axoneme (red) and the GFP::IFT dynein (green) respectively. Scale bars: 5 μm.

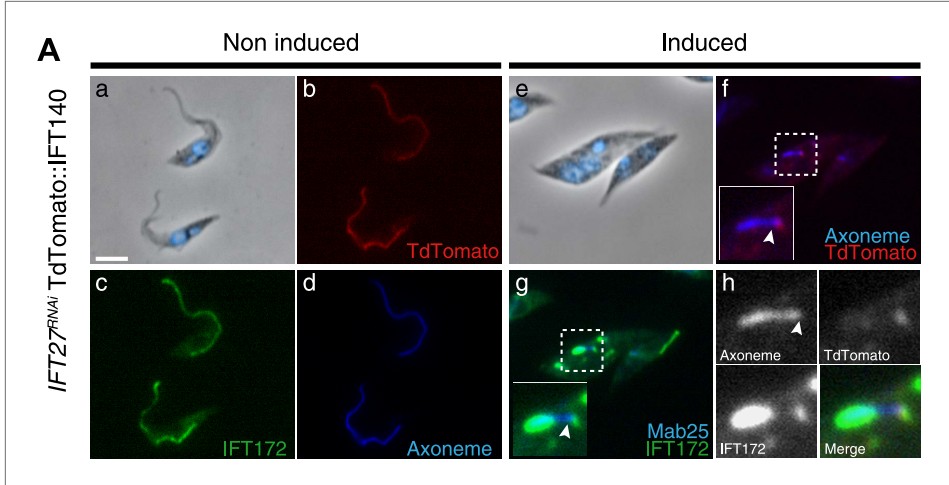

**Figure 8**. IFT140 does not accumulate in the short flagella of *IFT27^RNAi* cells. *IFT27^RNAi* cells were non-induced (a to d) or induced for 3 days (e to h) and simultaneously labeled with the anti-DsRed (red), the anti-IFT172 (green) antibodies along with Mab25 (blue) and counterstained with DAPI. Insets show the localization of the axoneme and TdTomato::IFT140 in f and of the axoneme and IFT172 in g. In h, the localization of Td::IFT140, IFT172 and the axoneme are shown. The arrowhead indicates the base of the flagellum. Scale bar: 5 μm.

Therefore, both actors of the retrograde transport fail to enter the flagellum in the absence of IFT27, suggesting that this protein, despite being a member of the IFT-B complex, is a major controller of retrograde transport.

## An IFT27 RNAi-resistant version complements the *IFT27^RNAi* cell line

To demonstrate the specificity of the *IFT27* silencing phenotype, an RNAi-resistant version of the gene was synthetized (GeneBank Accession Number KF147877) and expressed into the *IFT27^RNAi* cell line. Officially named *IFT27^RNAi+IFT27^RNAiRES* this new cell line will be termed *IFT27^RNAiRES* for simplicity. This exogenous copy of IFT27 has a C-terminal TY-1 tag (*Bastin et al., 1996*) to allow for independent monitoring. Upon induction of *IFT27* dsRNA expression, transcripts from the endogenous gene should be silenced, whereas the RNAi-resistant version of *IFT27* should remain unaffected. Western blot analysis of uninduced *IFT27^RNAiRES* cells using the anti-TY1 antibody showed a single band migrating close to the 20-kDa marker, whereas probing the membrane with the anti-IFT27 antibody revealed two bands, the endogenous IFT27 and the slightly larger exogenous IFT27^RNAiRES (the TY1-tagged protein has a predicted molecular weight of 22.64 kDa) (*Figure 9A*). The inhibition of the endogenous IFT27 occurred when RNAi was induced while the exogenous IFT27^RNAiRES continued to be expressed, as was evident in Western blots probed with the anti-IFT27 antibody (*Figure 9B*, left panel). Furthermore, IFT27 inhibition can be reversed after removing tetracycline, whereas IFT27^RNAiRES expression remained unchanged (*Figure 9B*, right panel). The growth curve of the *IFT27^RNAiRES* cell line showed no defects after RNAi induction (*Figure 9C*), and immunofluorescence assays using the anti-Ty1 antibody showed a signal all along the flagellum of both non-induced and induced cells, as expected (*Figure 9D*). These results show that the RNAi-resistant version of *IFT27* complements gene function upon silencing of the endogenous *IFT27* gene copies and demonstrate the specificity of the phenotypes observed in the *IFT27^RNAi* cell line.

To evaluate the ability of IFT27^RNAiRES to participate in IFT, a version fused to YFP at the C-terminus was synthetized and transfected into the *IFT27^RNAi* cell line in order to exclusively track the fluorescent RNAi-resistant version of IFT27 (*Figure 10*). Immunoblotting of this *IFT27^RNAiRES^YFP* cell line using the anti-IFT27 antibody showed reduced expression of endogenous IFT27 upon induction, while expression of IFT27^RNAiRES::YFP seems unaffected (*Figure 10A*). Live microscopy analyses revealed a clear localization of the fusion protein in the flagellum and at the base of the organelle regardless of RNAi induction (*Figure 10B*). As expected, the tracks made in the induced *IFT27^RNAiRES^YFP* cell line context are more noticeable than in the non-induced one. Kymograph analyses of IFT27^RNAiRES::YFP (*Figure 10C*; *Video 2*; *Figure 3*) revealed an average anterograde velocity of 2.3 ± 0.87 μm/s (n = 161)

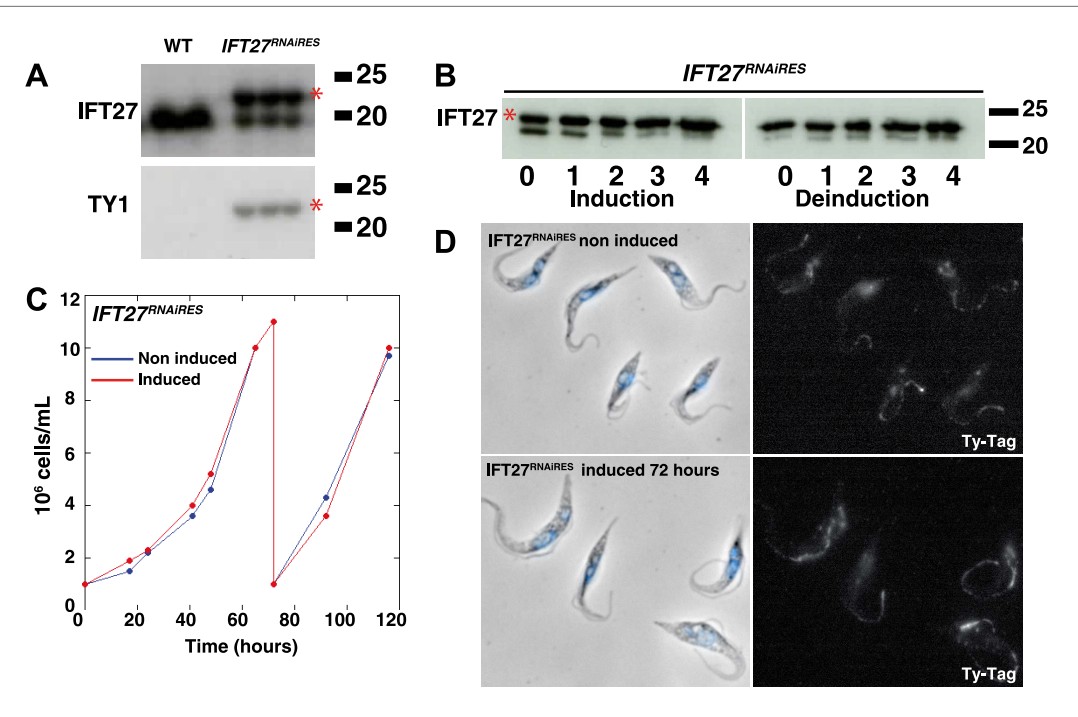

**Figure 9**. An epitope-tagged RNAi-resistant version of IFT27 complements the phenotype resulting from endogenous IFT27 depletion in the *IFT27^RNAi* cell line. (**A**) Western blot of total protein samples of wild-type (WT) and non-induced *IFT27^RNAiRES* cells. The anti-IFT27 antibody was used first and the anti-TY1 antibody was used after membrane stripping. The epitope-tagged RNAi-resistant version of IFT27 is indicated by the red asterisk. (**B**) Western blot of total protein samples from the *IFT27^RNAiRES* cell line prepared from induced cells and from cells deinduced after the indicated number of days. The red asterisk shows the presence of IFT27^RNAiRES. (**C**) Growth curves of the non-induced (blue) and induced (red) *IFT27^RNAiRES* cells. (**D**) Immunofluorescence of *IFT27^RNAiRES* cells using the anti-TY1 antibody. The first panel shows the phase-contrast image merged with DAPI staining and the second shows the BB2 anti Ty-1 antibody staining. Scale bar: 5 μm.

for the non-induced cells and 2.0 ± 0.53 μm/s (n = 177) for the induced cells. Retrograde IFT mean speed was measured at 4.4 ± 1.3 μm/s (n = 211) for non-induced cells and 4.0 ± 1.4 μm/s for induced cells (n = 227), values comparable with GFP::IFT27 (*Figure 2D*). Furthermore, immunofluorescence assays showed a clear colocalization of the endogenous and the recoded fusion protein in the flagellum and at the base of the organelle in both non-induced and induced context (*Figure 3B,C*). Overall, these results demonstrate the specificity of the *IFT27* silencing phenotype since the RNAi-resistant version of IFT27 complements the *IFT27^RNAi* phenotype, indicating that IFT27^RNAiRES and the YFP-tagged version of the protein are functional.

## A point mutation in the G1 domain of IFT27 impairs flagellum assembly

Rab proteins function as molecular switches cycling between a GTP-bound 'ON' form and a GDP-bound 'OFF' form. While multiple effector proteins recognize the GTP-bound form, the GDP-bound form is unable to form such interactions. These two conformational changes also control the cellular localization of Rab proteins (*Stenmark, 2009*) and as a consequence, mutations leaving the proteins in a GTP-locked or GDP-locked state affect the function and localization of this GTPases. To evaluate the significance of the GTP/GDP state of IFT27, two missense mutations were introduced at critical residues (*Figure 1*, GeneBank Accession Numbers KF147878 and KF147879) of the epitope-tagged *IFT27^RNAiRES* gene. The Q67L mutation replaces the catalytic glutamine at the G3 region found in most small G proteins with a leucine and is predicted to disrupt GTP hydrolysis, leaving the protein in a GTP-bound, active state. On the other hand, the mutation on the G1 domain replacing a threonine with an asparagine, T19N, is predicted to abolish GTP binding and should leave the protein in a GDP-bound, inactive state. These mutated versions were incorporated into the genome of the *IFT27^RNAi* cell line

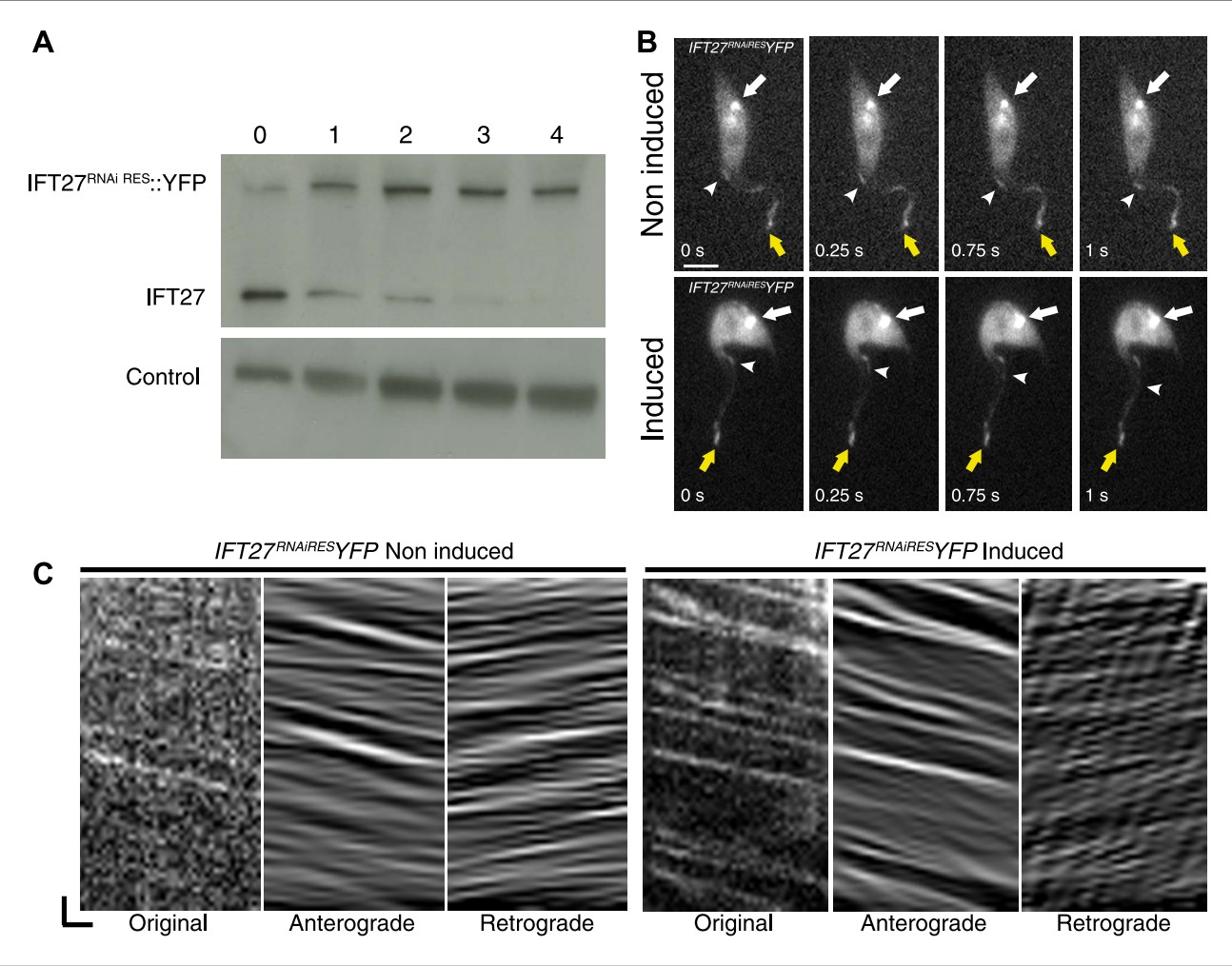

**Figure 10**. IFT27$^{RNAiRES}$::YFP traffics inside the trypanosome flagellum in the presence and absence of endogenous IFT27. (**A**) Western blot using the anti-IFT27 showing a decrease of the IFT27 endogenous protein. The anti-PFR L13D6 antibody was used as loading control and cells were induced for the indicated number of days. (**B**) Live microscopy imaging of non-induced (top panels) and induced (bottom panels) IFT27$^{RNAiRES}$YFP cells. White arrows show the fluorescent protein pool at the base of the flagellum and yellow arrows show the distal tip of the flagellum. Arrowheads indicate the successive position of an anterograde IFT train. (**C**) Kymographs show clear IFT tracks made by IFT27$^{RNAiRES}$::YFP in non-induced (left) and 3 day induced (right) cells (**Video 2**). Scale bars: 5 µm in **B**. In **C**, horizontal scale bar is 2 µm and vertical scale bar is 2 s.

context and were consequently named *IFT27$^{RNAi}$+IFT27$^{RNAiRES}$T19N* and *IFT27$^{RNAi}$+IFT27$^{RNAiRES}$Q67L* respectively. For simplicity, we will refer to them as *IFT27$^{RNAiRES}$T19N* and *IFT27$^{RNAiRES}$Q67L*.

Western blot showed that the amount of endogenous IFT27 decreased after 1 day of induction in both cell lines, leaving only the modified T19N **(GDP-locked)** and Q67L **(GTP-locked)** RNAi-resistant proteins (**Figure 11A,C**, stars). After induction, the growth rate of the *IFT27$^{RNAiRES}$Q67L* cell line did not vary (**Figure 11B**), while the *IFT27$^{RNAiRES}$T19N* cells showed impaired growth (**Figure 11D**). Therefore, the modified T19N protein produces a phenotype only when the endogenous IFT27 expression is inhibited, suggesting a recessive effect of the T19N mutation. Immunofluorescence assays using the monoclonal antibodies Mab25 and L8C4 showed that the *IFT27$^{RNAiRES}$Q67L* **(GTP-locked)** cells were undistinguishable from their non-induced control (**Figure 11E**) whereas *IFT27$^{RNAiRES}$T19N* **(GDP-locked)** cells build unusually short flagella after 3 days of induction (**Figure 11F**, arrows). The flagellar length of the *IFT27$^{RNAiRES}$Q67L* cell line showed no significant differences between non-induced (16.9 ± 1.9 µm, n = 100) and induced cells (16.6 ± 2.98 µm, n = 100) whereas flagellar length of *IFT27$^{RNAiRES}$T19N* cells dropped to 3.7 ± 3.1 µm (n = 79) after induction compared to 17.6 ± 2.1 µm (n = 100) for the non-induced controls. However, by carefully analysing both *IFT27$^{RNAiRES}$T19N* and *IFT27$^{RNAiRES}$Q67L* cells

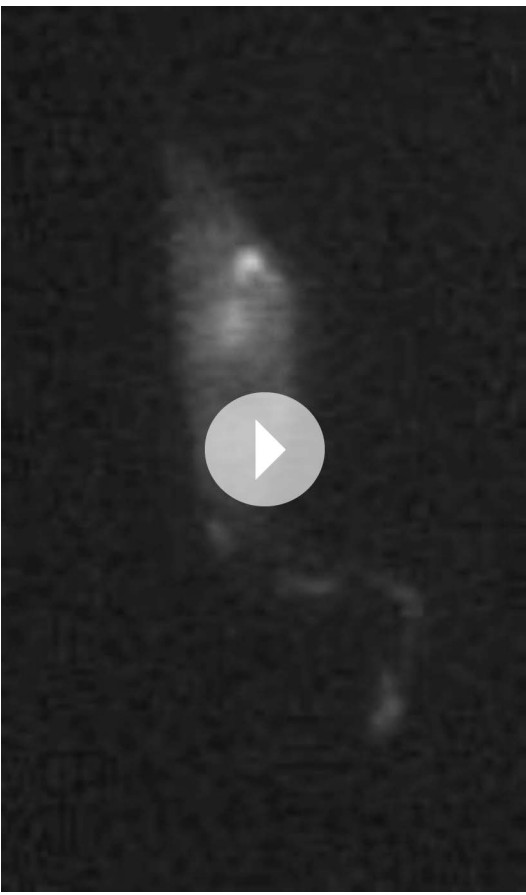

**Video 2**. IFT27**RNAiRES**::YFP traffics inside the trypanosome flagellum in the presence and the absence of endogenous IFT27. Observations of live, non-induced and 3-day-induced *IFT27^RNAiRES^YFP* cells using time-lapse epifluorescence microscopy using a DMI4000 microscope. Acquisitions were made at room temperature and frames were taken every 250 ms for 30 s by an Evolve 512 EMCCD Camera. Videos were exported to .AVI format with the ImageJ 1.47g13 software.

in non-induced conditions, we noticed that the length of the flagellum was shorter by about 3 μm compared to cell that are not expressing a mutated form of IFT27 (*Figure 12*). This interesting result indicates that the expression of a GDP or GTP-locked form of IFT27 has a slightly negative dominant impact on the formation of the trypanosome flagellum. Once the endogenous IFT27 is depleted, this effect is exacerbated in the case of the GDP-locked version but not in the case of the GTP-locked version (*Figure 12*). These observations reveal that the sole expression of the GTP-locked version, although able to sustain flagellum formation, is not neutral.

To investigate if the IFT machinery was disrupted in the *IFT27^RNAiRES^T19N* cell line, immunofluorescence assays using Mab25 and the anti-IFT172 were performed (*Figure 13A*). While the IFT172 staining looked normal in the non-induced context (*Figure 13Aa–d*), the induced *IFT27^RNAiRES^T19N* cells did not show signs of IFT B complex accumulation but IFT172 was found concentrated at the base of the organelle (*Figure 13Ae–h*) in a way reminding of the localization of the IFT140 protein in the *IFT27^RNAi^* cell line (*Figure 7*). This is in stark contrast with the *IFT27^RNAi^* cell line where the depletion of IFT27 is accompanied by accumulation of IFT172 inside the flagellum (*Figure 13B*). The phenotype obtained in the induced *IFT27^RNAiRES^T19N* cells is reminiscent to the ones observed upon disruption of the anterograde IFT machinery (*Pazour et al., 2000*; *Absalon et al., 2008b*) and shows that the expression of IFT27^RNAiRES^T19N in the absence of endogenous IFT27 triggers an anterograde IFT phenotype. To understand this intriguing phenotype, the flagellar structure of the *IFT27^RNAiRES^T19N* cell line was analyzed by TEM and scanning electron microscopy. TEM observations were not conclusive, as no visible defects in the axoneme structure

were detected. These results could be explained by the fact that *T. brucei* assembles the new flagellum while retaining the old one and that only this one, formed prior to the RNAi-mediated silencing of IFT27, could be observed. On the other hand, SEM revealed the presence of abnormally short flagella with no visible dilation (*Figure 13Aj*, arrow) and the presence of a 'flagellar sleeve', a long and thin extension of the flagellum membrane (*Figure 13Aj*, arrowheads). This is very different from the induced *IFT27^RNAi^* cells that possess a short, bulky flagellum, in agreement with the presence of an excessive accumulation of material (*Figure 12Bj*, arrow). Since the IFT dynein is found concentrated at the base of flagellum after IFT27 depletion (*Figure 8*), we wondered what could be the localization of the motor in the presence of the Q67L and T19N versions of IFT27. This was addressed by immunofluorescence assays using an anti-DHC1b antibody in non-induced and induced *IFT27^RNAiRES^Q67L* **(GTP-locked)** or *IFT27^RNAiRES^T19N* **(GDP-locked)** cells. In *IFT27^RNAiRES^Q67L* cells, the dynein signal showed no significant difference between the non-induced and induced conditions (*Figure 14*) suggesting that the motor is functional and participates normally to IFT. In contrast, DHC1b remained at the flagellum base in induced cells solely expressing the T19N mutant protein (*Figure 14e–h*, arrow). In conclusion, the GDP-locked version does not sustain the entry of the IFT-B complex in the flagellar

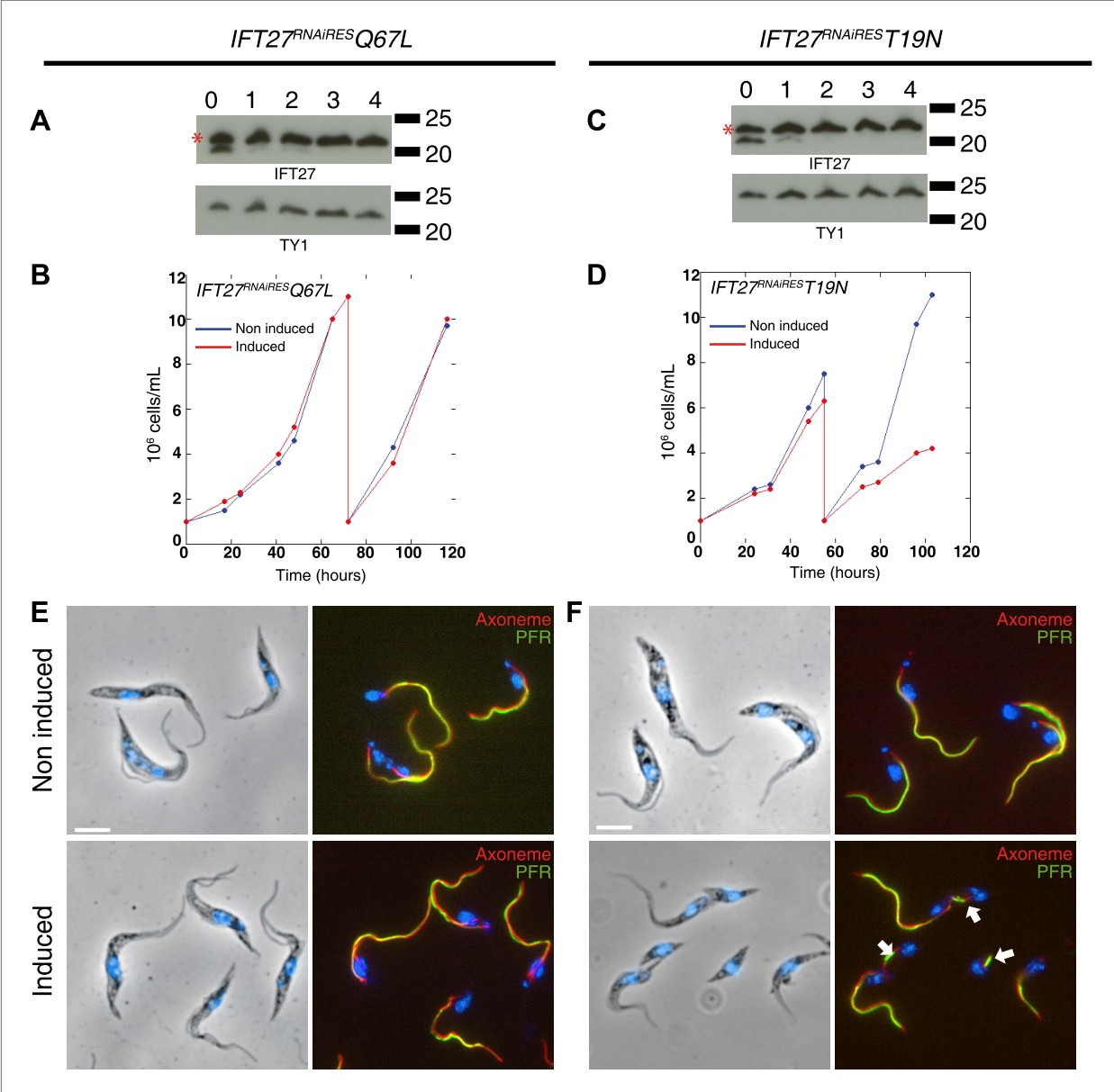

**Figure 11**. Expression of IFT27[RNAiRES]T19N and IFT27[RNAiRES]Q67L produces different phenotypes. (**A** and **C**) *IFT27[RNAiRES]Q67L* (**A**) and *IFT27[RNAiRES]T19N* cells (**C**) were induced the indicated number of days and western blots were performed using the anti-IFT27 antibody. A decrease in the amount of the endogenous protein can be observed while the expression of the modified RNAi-resistant proteins remains constant (red stars). (**B** and **D**) Growth curves of the non-induced and induced *IFT27[RNAiRES]Q67L* (**B**) and *IFT27[RNAiRES]T19N* (**D**) cell lines. (**E** and **F**) Immunofluorescence assays with the *IFT27[RNAiRES]Q67L* (**E**) and *IFT27[RNAiRES]T19N* (**F**) cells. After methanol fixation, non-induced and 3 day-induced cells were stained using L8C4 together with Mab25 to detect the PFR and the axoneme respectively. Arrows show abnormal flagella. Scale bar: 5 µm.

compartment, leading to an anterograde phenotype. Nevertheless, dynein is still targeted to the base of the flagellum.

## The modified T19N IFT27 protein does not access the flagellum

To evaluate the ability of the mutated versions of IFT27 to traffic inside the flagellar compartment, the IFT27[RNAiRES]T19N and Q67L point mutations were expressed as fusion proteins with YFP in the *IFT27[RNAi]* cell line. Immunofluorescence assays using the anti-IFT27 and the anti-GFP antibodies (*Figure 15*) showed that IFT27[RNAiRES]Q67L is localized inside the flagellum in non-induced (*Figure 15Aa–d*) and induced (*Figure 15Ae–h*) cells as the unmodified IFT27[RNAiRES]::YFP protein (*Figure 3*). In addition, the

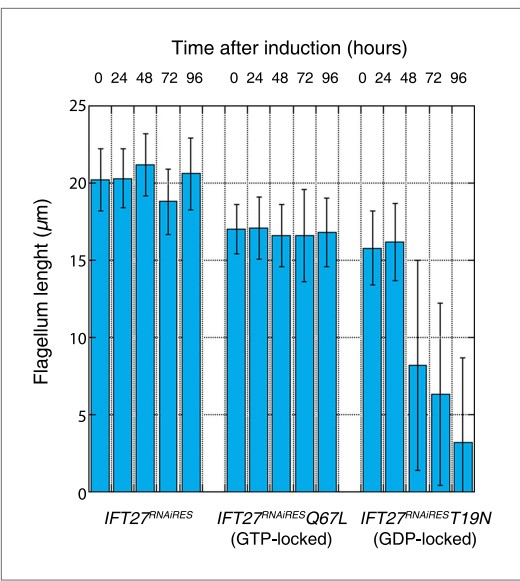

**Figure 12**. Expression of the GTP- or GDP-locked form of IFT27 has a slightly dominant-negative impact on the formation of the trypanosome flagellum. Flagellar length comparison of *IFT27*[RNAiRES] (expressing a non-mutant version of IFT27), *IFT27*[RNAiRES]*Q67L* (GTP-locked), and *IFT27*[RNAiRES]*T19N* (GDP-locked) cells. Flagellum length was measured after the indicated hours upon RNAi induction using the axoneme marker Mab25 (n = 200 for each induction time).

fluorescent Q67L modified protein traffics inside the flagellum in both conditions (**Figure 15B**; **Video 3**) and kymograph analyses demonstrated anterograde and retrograde train speed similar to the unmodified protein (**Figure 15B**, **Figure 3**).

Strikingly, the IFT27[RNAiRES]T19N::YFP fusion protein was always found in the cytoplasm in both non-induced and induced situations (**Figure 15C**). No signal could be detected in the flagellum, neither by immunofluorescence assays (**Figure 15C**) nor by live cell imaging (**Video 4**) in both non-induced and induced cells, suggesting that the modified protein is not able to access the flagellum. Since the induced *IFT27*[RNAiRES]*T19N* cell line shows no signs of IFT-B accumulation in contrast to the induced *IFT27*[RNAi] cell line (**Figure 12**), a possible explanation is that IFT27[RNAiRES]T19N disrupts the assembly of the IFT complex B.

To first confirm the interaction of IFT27 with the IFT complex B of *T. brucei*, co-immunoprecipitation assays were performed using the anti-GFP antibody. The IFT27[RNAiRES]::YFP and the two mutated proteins were successfully immunoprecipitated in all three cell lines as confirmed by western blot with the anti-IFT27 (**Figure 15D**, top panel). Next, immunoblotting using two antibodies recognizing IFT-B proteins IFT22 and IFT172 showed that the immune complexes formed by IFT27[RNAiRES]::YFP and IFT27[RNAiRES]Q67L::YFP contain both proteins (**Figure 15D**, left and middle rows). However, immunoprecipitation of IFT27[RNAiRES]T19N::YFP failed to bring down IFT22 and IFT172 (**Figure 15D**, right panel). These results demonstrate that IFT27[RNAiRES]Q67L is associated to the IFT-B complex like the non-modified protein, supporting the in vivo results (**Figure 15B**). In contrast, the T19N mutant version of IFT27 is unable to interact with at least two members of the IFT-B complex. Overall, these results demonstrate that IFT27 needs to be in a GTP-bound state to interact with the rest of the IFT complex and access the flagellar compartment.

## Discussion

Studies in different model organisms have shed light into the structure and nature of the IFT machinery but few data are available about the mechanisms coordinating the entry of the IFT complexes into the flagellum. Amongst the possible regulators, several small GTPases have been associated with the flagellum: ARL-13b is involved in the stabilization of ciliary assembly in *C. elegans* and its mutations are associated with a ciliopathy named Joubert syndrome (**Cantagrel et al., 2008**; **Cevik et al., 2010**; **Li et al., 2010**). The RAN importin system, known to be involved in nucleocytoplasmic transport, is responsible for the entry of the kinesin KIF17 inside the primary cilium (**Dishinger et al., 2010**). However, none of these proteins are part of any of the IFT complexes.

Here, we have shown by immunoprecipitation that IFT27 likely is a constitutive member of the IFT-B complex in *T. brucei*. Previous purification assays of the B complex using C3/DYF13 in *T. brucei* identified most of the IFT-B predicted proteins but were not able to detect IFT27, perhaps because of its short molecular weight and the under-representation of small polypeptides inherent to the MS analysis. Significantly, IFT20 and IFT25, the two other predicted IFT-B proteins of short size, were also missing (**Franklin and Ullu, 2010**). We observed that GFP-tagged IFT27 traffics in the flagellum in a similar way to IFT52 or IFT81, two well-studied IFT-B members (**Bhogaraju et al., 2013**; **Buisson et al., 2013**) showing that IFT27 likely associates to the IFT-B complex and participates to the IFT process (**Figure 16A**). These findings are consistent with the fact that *Chlamydomonas* IFT27 is part of IFT-B complex and traffics inside the flagellum (**Lucker et al., 2005**; **Qin et al., 2007**).

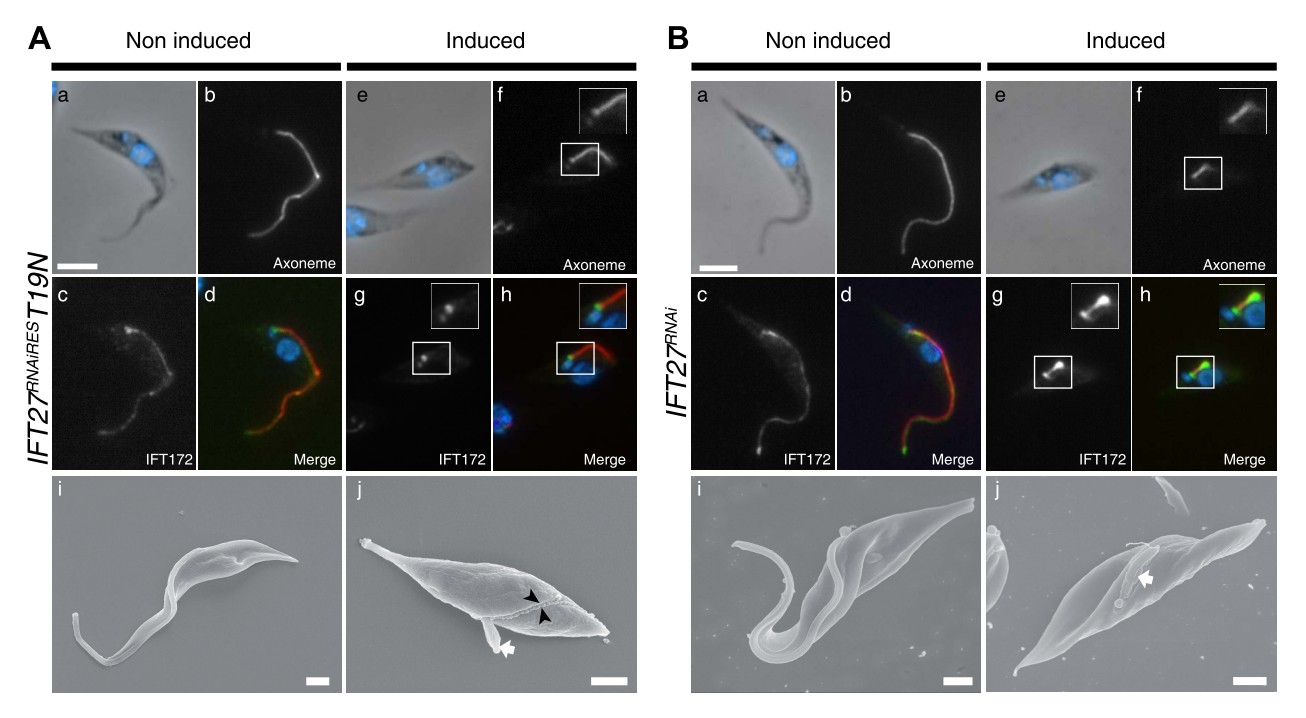

**Figure 13**. The *IFT27^RNAiRES^T19N* cell line displays an anterograde phenotype upon induction. (**A**) Non-induced (a to d) and 3-day-induced (e to h) *IFT27^RNAiRES^T19N* cells were fixed in methanol, stained with the anti-IFT172 and Mab25 and counterstained with DAPI. Insets show the axoneme and IFT172 localization. (i and j) Scanning electron microscopy of non-induced and induced *IFT27^RNAiRES^T19N* cells. The white arrow shows the short flagellum and the black arrowheads in j indicate the flagellar sleeve. (**B**) Immunofluorescence of non-induced (a to d) and 3-day-induced *IFT27^RNAi^* cells using the same conditions. Scanning electron microscopy of the non-induced and induced *IFT27^RNAi^* cell line. The white arrow in j points to the short flagellum with an abnormally large diameter. Scale bars: 5 μm from a to h, 1 μm in i and j.

Surprisingly for an IFT-B protein, IFT27 turns out to be essential for retrograde transport as its knockdown triggered the formation of short flagella with an unexpected accumulation of IFT-B proteins such as IFT52, IFT172, IFT22, and PIFTC3. By TEM, this accumulated material appeared as electron dense regions spread around short, irregular and often disrupted axonemes whereas the basal body and transition zone appeared normal. All these features are reminiscent of a retrograde phenotype, usually caused by a deficit of either the IFT dynein or the IFT-A complex (*Pazour et al., 1998*; *Ou et al., 2007*; *Absalon et al., 2008a*). Interestingly, DHC1b and IFT140, two essential molecules of the dynein and IFT-A complex respectively, are not able to enter the flagellum in the absence of IFT27 (*Figure 16B*). The IFT dynein is the retrograde motor of IFT that brings back IFT trains from the tip to the base of the flagellum, a process in which IFT-A is also involved although its exact contribution remains to be shown. Being an IFT-B protein, IFT27 could therefore regulate IFT-A and dynein transport at the physical interface between IFT-A, IFT-B and dynein complexes to ensure coordinate import in the flagellar compartment. Supporting that view, some relationships between IFT-B proteins and the IFT dynein motor have been suggested in the literature. First, IFT-B and dynein genes are both controlled at the transcriptional level by the transcription factor RFX whereas IFT-A and kinesin II are regulated differently in metazoans (*Thomas et al., 2010*). Second, co-immunoprecipitation experiments indicate that DHC1b interacts with the IFT-B protein IFT172 in *Chlamydomonas* (*Pedersen et al., 2006*). In *T. brucei*, we could not detect a direct interaction between IFT27 and DHC1b or IFT27 and IFT140 by co-immunoprecipitation (unpublished data) but the association might be transient, for example taking place only during protein entry in the flagellar compartment.

It has been shown that IFT27 forms a complex with IFT25, and both could be involved in flagellum entry regulation (*Wang et al., 2009*). In addition, the IFT25/27 structure displays a surface patch able to mediate protein–protein interactions (*Bhogaraju et al., 2011*). This IFT25/27 sub-complex could then act as a scaffold mediating the interaction of the IFT-B complex with dynein and IFT-A complexes.

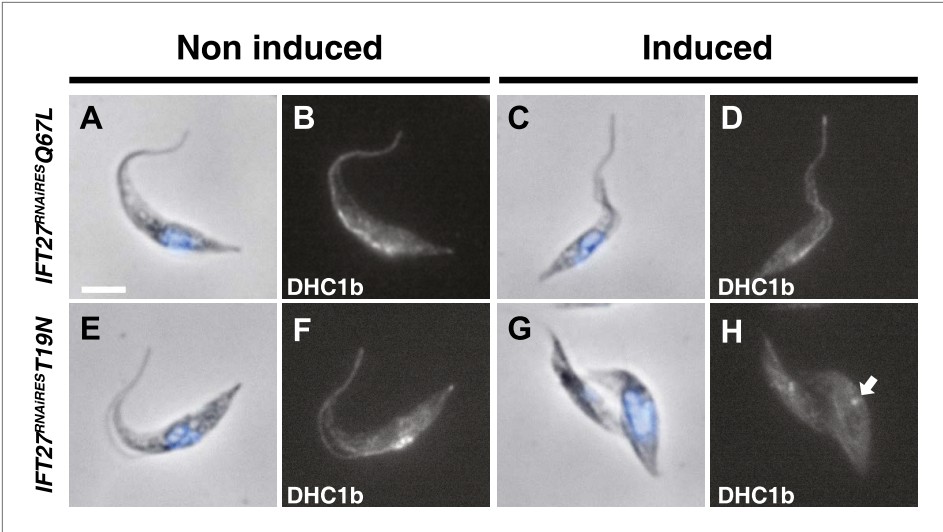

**Figure 14**. The *IFT27^{RNAiRES}T19N* cell line shows perturbation of IFT dynein entry into the flagellum. Non-induced and cells induced for 3 days of the *IFT27^{RNAi}Q67L* (a to d) and *IFT27^{RNAiRES}T19N* (e to h) cell lines were fixed in methanol, probed with the anti-DHC1b antibody and counterstained with DAPI. Arrows indicate the dynein pool around the base of the flagellum. Scale bar: 5 µm.

In *Chlamydomonas*, the RNAi knockdown of IFT27 produced cells with abnormally short flagella but the nature of the IFT defect (effect on anterograde or retrograde IFT) has not been reported (*Qin et al., 2007*). In this organism, it is the only IFT protein whose inhibition triggers cell cycle defects. However, the specificity of the phenotype has not been proven and an off-target effect cannot be excluded. In trypanosomes, the flagellum is essential for cell morphogenesis and cytokinesis and inhibition of all *IFT* genes studied so far invariably resulted in growth arrest (*Kohl et al., 2003*; *Davidge et al., 2006*; *Absalon et al., 2008a*; *Franklin and Ullu, 2010*). In the present study, the possibility of this being an off-target effect has been ruled out upon complementation of the *IFT27^{RNAi}* phenotype by the expression of an RNAi-resistant version of IFT27 that rescued both the observed growth and flagellar phenotype while displaying normal IFT trafficking in non-induced as well as in induced cells. IFT27 has all the necessary features to function as a GTPase, although it displays low intrinsic activity during in vitro assays (*Bhogaraju et al., 2011*). The orthologue of IFT27 in the related protozoan *Trypanosoma cruzi* exhibits GTPase activity in vitro (*Ramos et al., 2005*). As the *T. cruzi* protein is 88.5% similar to IFT27 and all the key residues are conserved, it is very likely that the *T. brucei* protein is also able to hydrolyze GTP. We have shown that the Q67L version apparently behaves like the unmodified protein even in the complete absence of the endogenous protein. Thus, IFT27 likely interacts with other IFT complex B members, controls the entry of the IFT-A complex and DHC1b and enters the flagellum in a GTP-bound state (*Figure 16A*). These findings could be compared with previous results showing that a constitutively active GTPase activity does not affect the localization of various Rab proteins (*Richardson et al., 1998*; *Alvarez et al., 2003*; *Nachury et al., 2007*). Another possible caveat to explain the absence of phenotype in the cells expressing the Q67L version of IFT27 is that the protein is not actually impaired in its GTPase activity so that it undergoes a normal chemical cycle. However, this is not likely the case as we noticed that cells expressing both the endogenous and the GTP-locked protein possessed a slightly shorter flagellum compared to cell that are not expressing a mutated form of IFT27 (*Figure 12*). These observations reveal that the sole expression of the GTP-locked version, although able to sustain flagellum formation, is not neutral.

Since the constitutively active version of IFT27 is found inside the flagellum and interacts with the rest of the IFT machinery, one could argue about the need of an inactive state. However, *T. brucei* has a complex life cycle in which the length of the flagellum can vary from 3 µm to 30 µm (*Rotureau et al., 2011*). Therefore, it will be interesting to see the impact of the GTP/GDP cycle on assembly of flagella of different length during the life cycle of *T. brucei*.

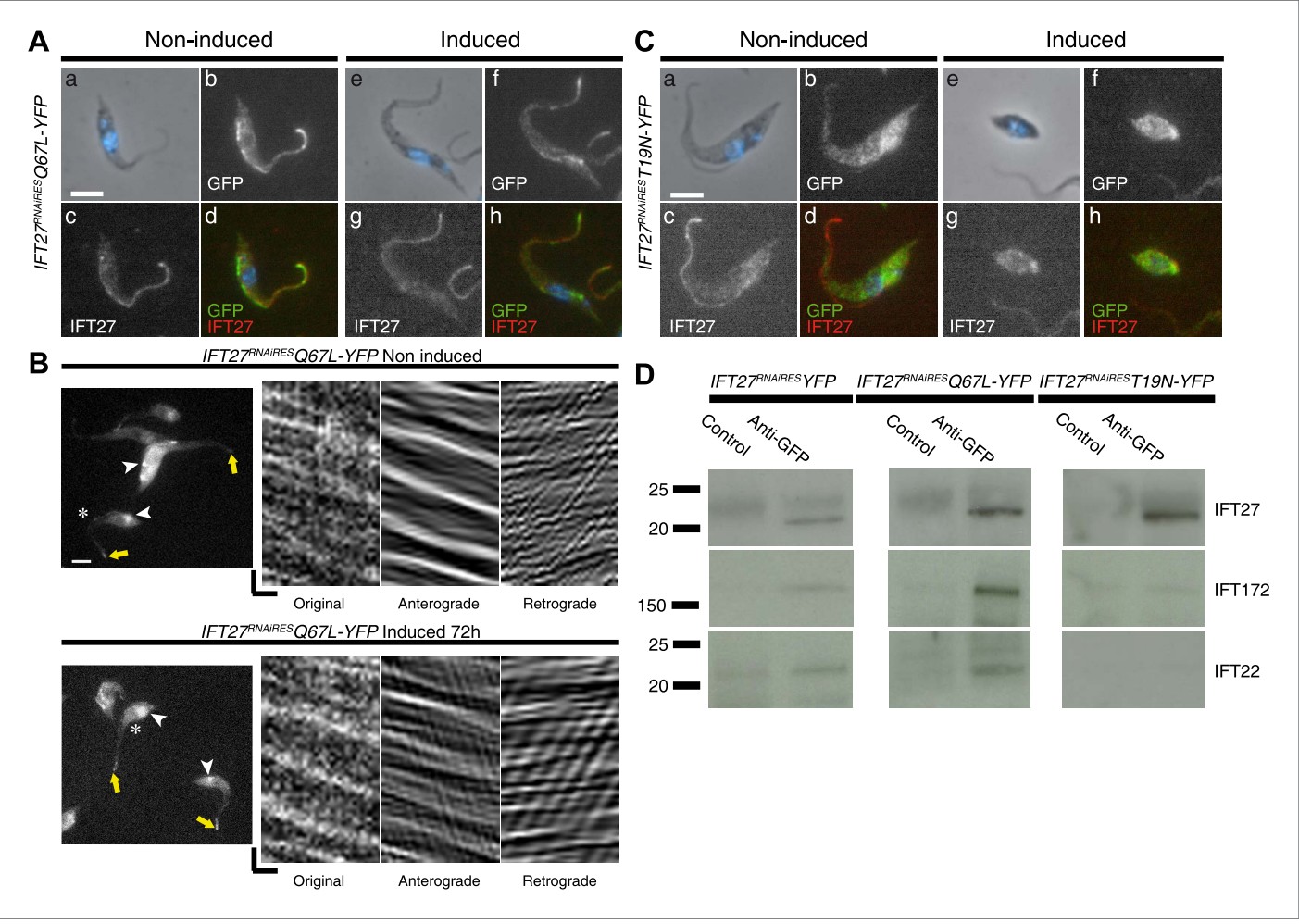

**Figure 15**. IFT27 in its inactive form is unable to penetrate the flagellar compartment. (**A**) *IFT27^RNAiRES^Q67LYFP* cells were non-induced (a to d) or induced for 3 days (e to h), stained with the anti-IFT27, the anti-GFP antibody and counterstained with DAPI. (**B**) Live observation of the non-induced and 3-day-induced *IFT27^RNAiRES^Q67LYFP* cell line. Yellow arrows indicate the flagellar tip, white arrowheads show the fluorescent protein pool at the base of the flagellum and the asterisk indicates the cells used for kymograph analysis. (**C**) Immunofluorescence assays of non-induced (a to d) or 3 day induced (e to h) *IFT27^RNAiRES^T19NYFP* cells using the anti-IFT27 and the anti-GFP antibody. Scale bar for (**A**), and (**C**): 5 µm. In (**B**), horizontal scale bar is 2 µm and vertical scale bar is 2 s. (**D**) Immunoprecipitates of total proteins by an anti-GFP antibody were separated on 4–15% polyacrylamide gels for the three different cell lines as indicated. Corresponding immunoblots with the anti-IFT172 (middle row) and the anti-IFT22 (bottom row) are shown and lanes labeled as 'Control' were precipitations with a rabbit antibody raised against *T. brucei* aldolase. These results were replicated in four independent experiments.

The expression of the T19N version, predicted to leave IFT27 in a GDP-bound form, did not cause a dominant negative phenotype. However, the modified T19N protein was not found inside the flagellar compartment neither in the presence nor in the absence of endogenous IFT27 and was unable to associate with the IFT-B complex. One would then expect that the exclusive expression of IFT27 in its inactive state should phenocopy the *IFT27^RNAi^* cell line. Strikingly, this was not the case as only very short flagella that do not exhibit accumulation of IFT proteins were built, typical of anterograde inhibition (**Absalon et al., 2008b**). Several explanations could be proposed to explain this intriguing result. First, the inactive IFT27 protein could sequester other components of the IFT-B complex in the cytoplasm and inhibit their transfer to the base of the flagellum. However, immunofluorescence assays revealed that IFT172 is still found concentrated at the base of the flagellum but not inside organelle. Second, the inactive protein could interfere with assembly or stability of the IFT-B complex in the cytoplasm, a hypothesis supported by the reported decrease in the amount of several IFT proteins in the *Chlamydomonas IFT27* RNAi experiments (**Qin et al., 2007**). However, this seems unlikely here

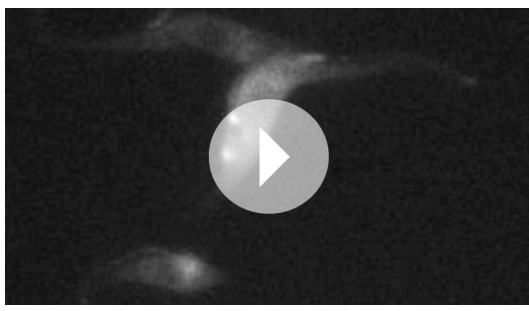

**Video 3**. IFT27 in its active form traffics normally in the flagellar compartment. Live, time-lapse epifluorescence microscopy of non-induced and 3-day induced *IFT27^{RNAiRES}Q67L::YFP*. All acquisitions were made at room temperature with frames taken every 250 ms for 30 s. Videos were exported to .AVI format with the ImageJ 1.47g13 software.

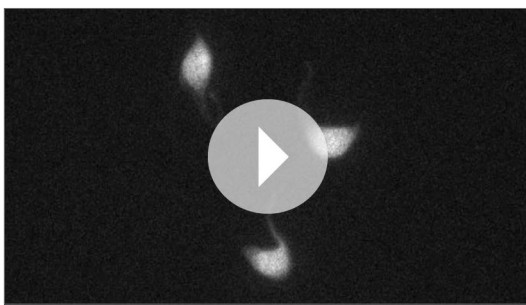

**Video 4**. IFT27 in its inactive form is unable to penetrate the flagellar compartment. Live, time-lapse epifluorescence microscopy of non-induced *IFT27^{RNAiRES}T19N::YFP* cells as well as 3-day induced ones. All acquisitions were made at room temperature with frames taken every 250 ms for 30 s. Videos were exported to .AVI format with the ImageJ 1.47g13 software.

because the total amount of IFT172 and IFT22 did not decrease when the IFT27T19N protein was exclusively expressed in the *IFT27^{RNAiRES}T19N* cell line (unpublished data). Third, the inactive IFT27 could negatively control the entry of the IFT-B complex hence resulting in an anterograde phenotype while the wild-type protein in non-induced conditions would overhaul this effect.

Curiously, both IFT27 and IFT25 are absent from *C. elegans*, *D. melanogaster*, and *Giardia intestinalis* (*van Dam et al., 2013*). This could be the consequence of different organization of the flagellum base, a sub-compartment involved in the regulation of the entry of flagellar components (*Reiter et al., 2012*). No obvious basal body structure can be found at the base of *C. elegans* cilia (*Perkins et al., 1986*; *Williams et al., 2011*) and in *D. melanogaster*, the basal body possesses a unique electron-dense ring structure (*Enjolras et al., 2012*). The δ- and ε-tubulins, involved in basal body maturation, are also absent from the genomes of these two organisms (*Dutcher, 2001*). Finally, the basal bodies are deeply rooted in the cytoplasm in *Giardia*, with long cytoplasmic axoneme segments (*Dawson and House, 2010*). Therefore, the absence of IFT27 and IFT25 could be related to the non-canonical structure of their flagellum base, implying a different process for the entry of the IFT proteins. Alternatively, the requirement for retrograde transport might be different according to the type of cilia, necessitating specific adaptations for IFT-A/IFT dynein trafficking. Inhibition of retrograde IFT moderately affects the length of sensory cilia in *Drosophila* (reduced by one third in IFT140 or dynein mutants, *Lee et al., 2008*) or *C. elegans* (reduced by 25% in IFT-A mutants, *Blacque et al., 2006*) in contrast to the severe reduction observed in *Chlamydomonas* (*Pazour et al., 1999*), trypanosomes (*Kohl et al., 2003*) or *Leishmania* (*Satir and Christensen, 2007*). In summary, these findings demonstrate that a GTPase-competent IFT27 is required for association to the IFT complex and that IFT27 plays a role in the cargo loading of the retrograde transport machinery.

## Materials and methods

### Plasmids, cell lines, and culture conditions

All procyclic *T. brucei* cell lines were derivatives of the strain 427 and grown in SDM79 medium with hemin and 10% fetal calf serum. The 29–13 cell line expressing the T7 RNA polymerase and the tetracycline-repressor (*Wirtz et al., 1999*) has been described previously, as well as the cell line expressing GFP::IFT52 (*Absalon et al., 2008b*). For generation of the *IFT27^{RNAi}* cell line, a 360-nucleotide fragment of *IFT27* (Tb 927.3.5550) was amplified by PCR and cloned in the pZJM vector (*Wang et al., 2000*), allowing tetracycline-inducible expression of dsRNA generating RNAi upon transfection in the 29-13 recipient cell line. The dsRNA is expressed from two tetracycline-inducible T7 promoters facing each other in the pZJM vector. Primers were selected using the RNAit algorithm to ensure that the fragment lacked significant identity to other genes to avoid cross-RNAi (*Redmond et al., 2003*). The fragment was amplified by PCR with the forward primer cgatcgAAGCTTaaacttgcgtcttcaggtgg and the

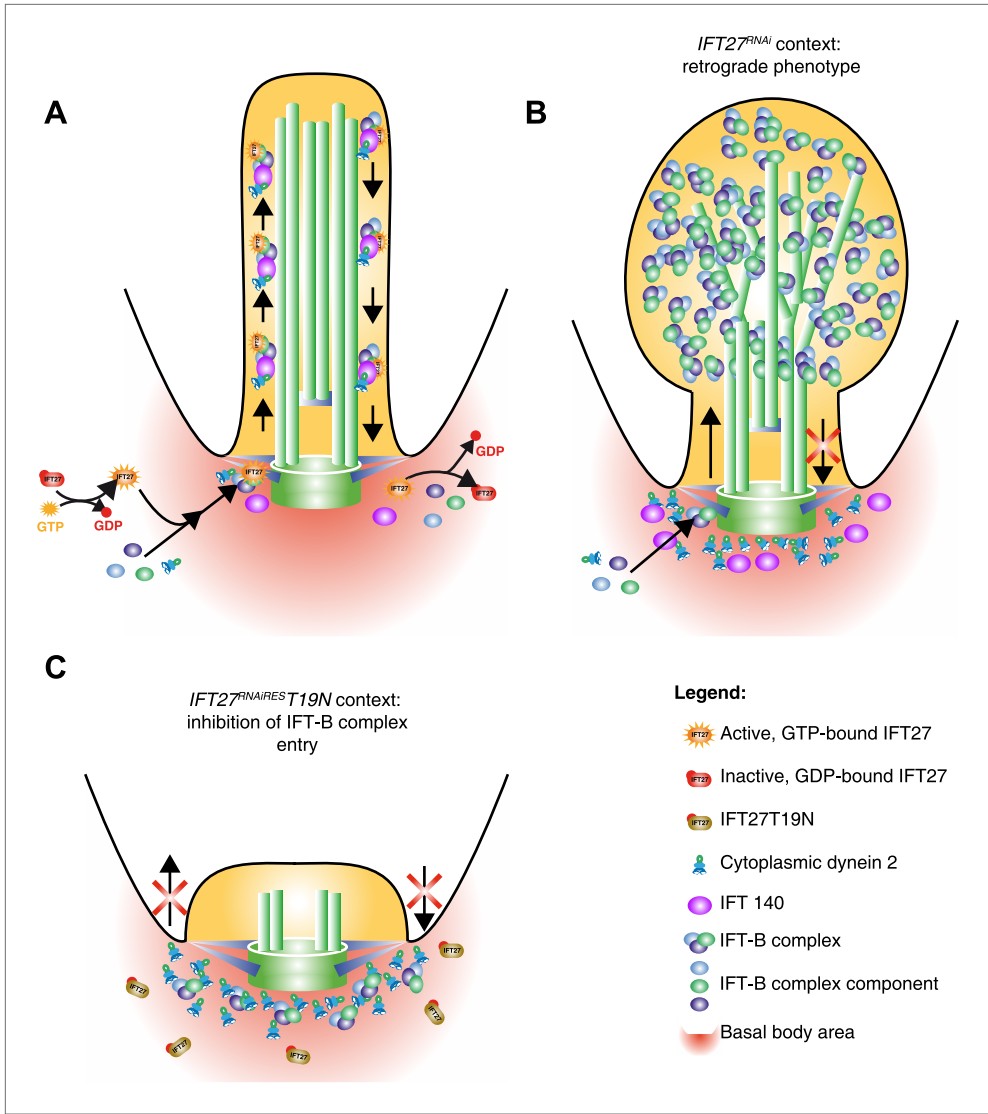

**Figure 16**. Model of IFT27 function during IFT assembly. (**A**) The proposed model shows a GTP-bound IFT27 associated with the IFT-B complex prior to the entry into the flagellum. After leaving the flagellar compartment, GTP is hydrolyzed, releasing IFT27 and disassembling the IFT-B complex. (**B**) After RNAi-mediated depletion of IFT27, the IFT-B complex enters the flagellum but is unable to exit because of the absence of IFT dynein inside the flagellar compartment. To explain the anterograde phenotype seen in after expression of the GDP-bound version of IFT27, the inactive IFT27 could negatively control the entry of the IFT-B complex (**C**). The PFR was not drawn for clarity.

reverse primer gtcat<u>CTCGAG</u>atttgcgagatctttcccct, digested with XhoI and HindIII (sites underlined) and ligated into the corresponding sites of the pZJM vector. The plasmid was linearized at the unique NotI site in the rDNA intergenic targeting region before transfection (**Wang et al., 2000**). GeneCust Europe (Dudelange, Luxembourg) conducted the chemical synthesis of the pPCPFReGFPIFT27 plasmid allowing a GFP tagging at the N-terminal of IFT27. The whole *IFT27* gene (552 nucleotides) was synthetized, digested with NheI and EcoRV and cloned into the corresponding sites of the pPCPFR vector (**Adhiambo et al., 2009**). The resulting plasmid was linearized with NsiI, targeting integration in the intergenic region of *PFR2*. Generation of the p2845TdTomatoIFT140 plasmid enabling the expression of TdTomato::IFT140 was performed by modifying the p2845IFT140 mCherry plasmid. The mCherry gene was removed from p2845IFT140 using XhoI and HindIII while simultaneasouly getting the TdTomato gene from the p2675TdTomatoIFT81 plasmid with the same restriction enzymes. Ligation was performed between the digested p2845IFT140 plasmid and the TdTomato insert

overnight at 16°C, the resulting plasmid was then transfected into XL-10 Gold Ultracomptent Cells (Agilent Technologies, France) to allow amplification. Plasmids from 10 different colonies were double digested with XhoI and HindIII to assess the presence of the TdTomato insert.

For the RNAi-resistant version of IFT27, a new version of the gene was chemically synthesized modifying the first 411 nucleotides in order to make synonymous mutations, leaving the amino acid sequence unchanged (GeneBank Accession Number KF147877). This IFT27RNAires gene was then cloned into the pPCPFR vector. Additional point mutations (*Figure 1*) made by GeneCust Europe enabled the creation of the RNAi-resistant versions of IFT27 in a GTP (Q67L) or GDP (T19N) locked state (GeneBank Accession Numbers KF147878 and KF147879). The YFP-tagged version of IFT27RNAires was made by synthetizing and cloning the *YFP* gene into the pPCPFRIFT27RNAires vector between the NheI and BamHI restriction sites. The Q67L and T19N point mutations were then performed to finally obtain the C-terminus-tagged *pPCPFRIFT27^{RNAires}Q67LYFP* and *pPCPFRIFT27^{RNAires}T19NYFP* plasmids. Linearization was carried out with NsiI. In all cases, trypanosomes were transfected with the plasmid constructs by Nucleofector technology (Lonza, Italy) as described before (*Burkard et al., 2007*). Transfectants were grown in media with the appropriate antibiotic concentration and clonal populations were obtained by limited dilution.

## Protein expression and antibody production

For antibody production, IFT27 was expressed as a recombinant fusion GST protein in *Escherichia coli*. The full length *IFT27* coding sequence was amplified by PCR using Platinum *Taq* Polymerase High Fidelity (Invitrogen) and primer pairs gaggaaGGATCCatggtaaacttgcgtcttcagg and cgcaacagCCCGGGt-tacctcatttggg (BamHI and XmaI sites underlined). PCR products were ligated into the pCR2.1-TOPO vector (Invitrogen, Carlsbad, CA) following the manufacturer's instructions. *IFT27* sequence was then excised using *Bam*HI and *Xma*I enzymes and ligated into compatible sites of the pGEX B vector (GE Healthcare, Piscataway, NJ). The plasmid was sequenced to confirm correct fusion with GST and transformed in *E. coli* BL21. Protein expression was triggered by adding 0.1 mM IPTG for 3 hr at 37°C and analyzed by SDS-polyacrylamide gel electrophoresis (PAGE) followed by Coomassie staining. Glutathione transferase (GST)-coupled proteins were purified as described previously (*Smith and Johnson, 1988*) and 20 μg were administrated by four successive subcutaneous injections (every 3 weeks) to five BALB/c mice for immunization. After bleeding, sera were absorbed against GST. Sera from mice immunized with GST alone were used as negative controls.

## Transmission and scanning electron microscopy

For transmission electron microscopy, cells were fixed in culture media overnight at 4°C in 2.5% gluta-raldehyde–0.1 M cacodylate buffer (pH 7.2) and postfixed in $OsO_4$ (2%) in the same buffer. After serial dehydration samples were embedded in Agar 100 (Agar Scientific, Ltd., United Kingdom) and left to polymerize at 60°C. Ultrathin sections (50–70 nm thick) were collected on Formvar-carbon-coated copper grids using a Leica EM UC6 ultramicrotome and stained with uranyl acetate and lead citrate. Observations were made on a Tecnai 10 electron microscope (FEI) and images were captured with a MegaView II camera and processed with AnalySIS and Adobe Photoshop CS4 (San Jose, CA). For scanning electron microscopy, samples were fixed overnight at 4°C in 2.5% glutaraldehyde–0.1 M cacodylate buffer (pH 7.2) and postfixed in 2% $OsO_4$ in the same buffer. After serial dehydration, the samples were dried at the critical point and coated with platinum according to standard procedures. Observations were made in a JEOL 7600F microscope.

## Immunofluorescence and live cell imaging

Cultured parasites were washed twice in SDM79 medium without serum and spread directly onto poly-L-lysine coated slides. The slides were air-dried for 10 min, fixed in methanol at −20°C for 30 s and rehydrated for 10 min in PBS. For immunodetection, slides were incubated with primary antibodies diluted in PBS with 0.1% Bovine Serum Albumine (BSA) for 1 hr. Three washes of 10 min were performed and the secondary antibody diluted in PBS with 0.1% BSA was added to the slides. After an incubation of 45 min, slides were washed three times in PBS for 10 min and DAPI (2 μg/μl) was added. Slides were mounted using ProLong antifade reagent (Invitrogen).

Antibodies used were the anti-IFT27 antisera of one mouse diluted 1/800, the anti-IFT22/RABL5 mouse polyclonal antibody (*Adhiambo et al., 2009*), Mab25 recognizing TbSAXO1, a protein found all along the trypanosome axoneme (*Pradel et al., 2006*), L8C4, recognizing PFR2, a major paraflagellar rod protein (*Kohl et al., 1999*), BB2 recognizing the TY1-epitope (*Bastin et al., 1996*), the anti-IFT172 mouse monoclonal and anti-DHC1b polyclonal antibodies diluted 1/200 and 1/100 respectively (our unpublished

data), the Living Colors DsRed Polyclonal Antibody (Clontech) diluted 1/1500 and the rabbit polyclonal anti-GFP (Invitrogen) diluted 1/500. Subclass-specific secondary antibodies coupled to Alexa 488 and Cy3 (1/400; Jackson ImmunoResearch Laboratories, West Grove, PA) were used for double labeling. Sample observation was performed using a DMI4000 microscope equipped with a 100X NA 1.4 lens (Leica, Wetzlar, Germany) and images captured with a CoolSnap HQ camera (Roper Scientific, Tucson, AZ). Pictures were analyzed using ImageJ 1.47g13 software (National Institutes of Health, Bethesda, MD) and images were merged and superimposed using Adobe Photoshop CS4. For live video microscopy, cells were covered by a coverslip and observed directly with the DMI4000 microscope at room temperature. Videos were acquired using an Evolve 512 EMCCD Camera (Photometrics, Tucson, AZ), the Metamorph acquisition software (Molecular Probes, Sunnyvale, CA). Trypanosomes were observed and the IFT was recorded at 250 ms per frame during 30 s using the Evolve 512 EMCCD Camera. Videos were mounted using iMovie 2011. Kymographs were extracted and analyzed as described previously (*Chenouard et al., 2010*, *Buisson et al., 2013*).

## Western blot

Cells were washed in PBS and boiled in Laemmli loading buffer before SDS-PAGE separation, loading 20 µg of total cell protein per lane. Proteins were transferred overnight at 25V at 4°C to polyvinylidene fluoride membranes, then blocked with 5% skimmed milk in PBS-Tween 0.1% (PBST) and incubated with primary antibodies diluted in 1% milk and PBST. The anti-IFT27 serum was diluted 1/800, and BB2 was diluted 1/100. To detect GFP and YFP, we used a mouse monoclonal anti-GFP antibody (Roche) diluted 1/500. As loading controls, antibodies anti-PFR (L13D6) (*Kohl et al., 1999*) diluted 1/50 and anti-aldolase (a kind gift of Paul Michels, Brussels, Belgium) diluted 1/1000 were used. Three membrane washes were performed with PBST for 5 min. Species-specific secondary antibodies coupled to horseradish peroxidase (GE Healthcare) were diluted 1/20,000 in PBST containing 1% milk and incubated for 1 hr. Final detection was carried out by using an enhanced chemoluminescence kit and a high performance chemoluminescence film according to manufacturer's instructions (Amersham, Piscataway, NJ).

## Immunoprecipitations

50 ml of *IFT27^RNAiRES^YFP*, *IFT27^RNAiRES^Q67LYFP* and I*FT27^RNAiRES^T19NYFP* cell lines were grown until they reached a concentration of $1.10^7$ cells/ml. Cells were washed and incubated in $IP_2$ buffer (0.8M NaCl, 20 mM Tris–HCl pH 8, 20 mM EDTA, 2% Triton X100, 10 µg/ml leupeptin and pepstatine A) for 30 min at room temperature. Soluble protein extract was centrifuged at 21,130×$g$ at 4°C for 30 min and supernatant was collected and incubated for 12 hr at 4°C with either 5 µl of polyclonal rabbit anti-GFP (Invitrogen) or 5 µl of anti-aldolase in the control. Prior to the immune complex recovery, protein A-Sepharose beads (GE Healthcare, NJ) were washed in $IP_1$ buffer (0.4M NaCl, 10 mM Tris–HCl pH 7, 4, 10 mM EDTA, 1% Triton X100, 1 mg/ml ovalbumine). Immune complexes were recovered by incubation with pretreated protein A-Sepharose beads for 15 min at 4°C. The beads then were washed three times with $IP_1$ buffer, twice with $IP_2$ buffer and once with $IP_3$ buffer (0.4M NaCl, 10 mM Tris–HCl pH 7,4, 10 mM EDTA). Interaction of the immune complex with the beads was disrupted by boiling in 50 µl of Laemmli loading buffer and analyzed by SDS-PAGE followed by immunoblotting.

## Acknowledgements

We acknowledge Derrick Robinson (Bordeaux, France) and Paul Michels (Brussels, Belgium) for generous gifts of monoclonal antibodies. We are thankful to the Plateforme de Microscopie Ultrastructurale for providing access to their equipment. We also thank Brice Rotureau, Benjamin Morga, and Cher-Pheng Ooi for critical reading of the manuscript.

## Additional information

### Funding

| Funder | Grant reference number | Author |
| --- | --- | --- |
| Institut Pasteur | | Diego Huet, Thierry Blisnick, Sylvie Perrot, Philippe Bastin |
| Centre national de la recherche scientifique | | Diego Huet, Thierry Blisnick, Sylvie Perrot, Philippe Bastin |

| Funder | Grant reference number | Author |
| --- | --- | --- |
| ANR | 11-BSV8-016 | Diego Huet, Thierry Blisnick, Sylvie Perrot, Philippe Bastin |
| Fondation pour la Recherche Médicale (FRM) | | Diego Huet |
| French National Ministry of Research and Technology | | Diego Huet |

The funders had no role in study design, data collection and interpretation, or the decision to submit the work for publication.

## Author contributions

DH, Conception and design, Acquisition of data, Analysis and interpretation of data, Drafting or revising the article; TB, Acquisition of data, Drafting or revising the article, Contributed unpublished essential data or reagents; SP, Acquisition of data, Analysis and interpretation of data; PB, Conception and design, Analysis and interpretation of data, Drafting or revising the article

# Additional files

## Major datasets

The following datasets were generated:

| Author(s) | Year | Dataset title | Dataset ID and/or URL | Database, license, and accessibility information |
| --- | --- | --- | --- | --- |
| Huet D, Perrot S, Blisnick T, Bastin P | 2013 | Synthetic construct clone DH-G086 IFT27 (IFT27) gene, complete cds | https://www.ncbi.nlm.nih.gov/nuccore/KF147877 | Publicly available at NCBI GenBank (https://www.ncbi.nlm.nih.gov/nuccore). |
| Huet D, Perrot S, Blisnick T, Bastin P | 2013 | Synthetic construct clone DH-G087 IFT27 (IFT27) gene, complete cds | https://www.ncbi.nlm.nih.gov/nuccore/KF147878 | Publicly available at NCBI GenBank (https://www.ncbi.nlm.nih.gov/nuccore). |
| Huet D, Perrot S, Blisnick T, Bastin P | 2013 | Synthetic construct clone DH-G089 IFT27 (IFT27) gene, complete cds | https://www.ncbi.nlm.nih.gov/nuccore/KF147879 | Publicly available at NCBI GenBank (https://www.ncbi.nlm.nih.gov/nuccore). |

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
