## [Decision Letter]

Thank you for sending your work entitled “The GTPase IFT27 participates in the regulation of anterograde and retrograde intraflagellar transport” for consideration at *eLife*. Your article has been favorably evaluated by a Senior editor, a Reviewing editor, and 3 reviewers.

The Reviewing editor and the reviewers discussed their comments before we reached this decision, and the Reviewing editor has assembled the following comments to help you prepare a revised submission.

Three reviewers read your paper, and after discussions with the editor, they recommend publication of the work. The main discussions centered around the biological insight. Those reviewers further from the field had a harder time understanding the insight than those closer to the field. Therefore we would like you to reemphasize the biological discovery here, which is not the fact that IFT27 is an important player in delivering cargo to the flagella, but rather the specific cargo is actually the retrograde machinery. Please make this clear in your Abstract, Introduction, and Discussion.

Experimentally, the major concern is the surprising and actually quite striking lack of kymograph analysis of anterograde and retrograde trafficking for any protein other than IFT27 itself. At least the most naive expectation would be that if IFT27 affects retrograde but not anterograde transport, then this should be evident as a drop in the retrograde transport frequency without a corresponding drop in the anterograde frequency. We do agree that their end-point phenotypic analysis, showing accumulation of IFTB proteins in short swollen cilia full of electron dense material does strongly support a specific defect in retrograde transport, but wouldn't a live cell analysis make this point much more directly? Perhaps there are tricky timing issues: obviously one would have to do the analysis early enough in the RNAi process that the cilia had not yet become short stumps. The authors do mention that such data exists (for example the authors mention mention having done this analysis for IFT140) but they do not show it. In particular, IFT52 kymographs would strongly support their model of impaired retrograde return being the cause for IFTB accumulation in the RNAi lines. You may have had good reasons why this analysis wasn't possible but I think that you should at least spell those reasons out, since this really seemed like a glaring omission, especially given the incredibly thorough level of treatment that the authors gave to every other aspect of their story.

Another issue that one of the reviewers brought up was the fact that the GTP locked form of IFT27 can fully replace the function of wild-type IFT27 (if I am understanding the results correctly). This is surprising since one normally thinks of G proteins as executing a cyclic reaction so that jamming the reaction at any point in the cycle should impair function. I bring this up because it was a surprising aspect of the story, but to their great credit the authors face this surprise head on and give several possible explanations. These seem reasonable but worth commenting that this was the one point in the paper where the story suddenly deviated from what one might have expected. Perhaps there is an explanation. IFT is a cycle, with anterograde particles switching to retrograde motion and then re-entering the flagella for another anterograde run. Given the cyclic nature of IFT, and the fact that IFT27 is involved in coupling retrograde and anterograde motion, a palausible model could have been that the GTPase cycle of IFT27 was coupled to the directional cycle of IFT. There was never really any reason to think this, but many readers will make a similar assumption and thus be startled by the active site mutant results. We would thus encourage the authors to emphasize a bit more explicitly the idea that the two cycles might have been coupled but apparently are not. That may help readers to avoid confusion.

---

## [Author Response]

Two major points were raised by the Reviewing editor:

*1) The lack of kymograph analysis of anterograde and retrograde trafficking for any protein other than IFT27 itself*.

This is an interesting suggestion as it could directly show the role of IFT27 in retrograde transport. We indeed considered doing this experiment in an *IFT27*^*RNAi*^ cell line expressing another IFT protein fused to a fluorescent protein, using our well set-up procedure to extract kymographs. However, we did not do these experiments because a similar approach using the *IFT140*^*RNAi*^ strain, a reference strain for inhibition of retrograde transport, turned out to be inconclusive (Fort & Bastin, unpublished data). In this case, the reporter protein was a fusion between YFP and IFT81, a core protein of the IFT-B complex (Bhogaraju et al, Science 2013). Two obstacles probably explain this. First, when cells were analysed at early time points of RNAi when they still possess a flagellum of reasonable length, no statistically significant differences could be found in the ratio between anterograde and retrograde trains (Slide 1, Films 19h and 25h). This could be due to the fact that in these cells, the unbalance is too discrete to be detected. One has to keep in mind that we can monitor IFT only for a few minutes before fading of the GFP/YFP signal whereas it takes 4-5 hours to assemble a full-length flagellum.

Therefore, a slight unbalance might hence go unnoticed over a short period but could have consequences on the actual amount of IFT-B proteins in the flagellum. Second, as time progresses, the number of cells with a flagellum that can be analyzed drops due to the rapid emergence of cells with a new tiny flagellum filled with IFT particles. In the few cells that can technically be analyzed, the amount of IFT material is high, turning the flagellum very bright and preventing observation of individual trains (Slide 1, Film 49h). Since we could not reach a clear conclusion on the seminal retrograde transport mutant, we did not try the experiment on the *IFT27*^*RNAi*^ strain. Nevertheless, as pointed out by the reviewers, we think that the end-stage phenotype of piling-up of IFT proteins in the short flagellum is demonstrating convincingly the involvement of IFT27 in retrograde IFT. This phenotype has been taken as landmark for attributing a retrograde function to multiple genes in for example *Chlamydomonas* (Pazour et al JCB1999), *C. elegans* (Blacque et al MBC2006) or humans (Perrault et al AJHG2012).

We examined the trafficking of TdTomato::IFT140 and of GFP::DHC2.1 in the *IFT27*^*RNAi*^ strain and could not detect any unbalance between anterograde and retrograde transport. However, this was somehow expected since the defect in that case is due to failure of entry of IFT-A and dynein motors. We did not show the film since a video where nothing moves is not very informative. The distribution of these two proteins is shown by immunofluorescence in Figures 7 and 8.

*2) The fact that the GTP locked form of IFT27 can fully replace the function of wild- type IFT27*.

We wish to thank the reviewers for this important question. By carefully analyzing the GDP and the GTP-locked mutants in conditions where the endogenous IFT27 protein is still expressed, we noticed that the length of the flagellum was shorter by about 3μm. This indicates that the expression of the GDP- or the GTP-locked versions has a slightly dominant negative impact on the formation of the flagellum. Once the endogenous IFT27 is depleted, this effect is exacerbated in the case of the GDP-locked version but not in the case of the GTP-locked version. These results reveal that the sole expression of the GTP-locked version, although able to sustain flagellum formation, is not neutral. A new figure has been introduced to describe this result and a new section has been added to the text.